# Eye features and retinal photoreceptors of the nocturnal aardvark (*Orycteropus afer*, *Tubulidentata*)

Leo Peichl[1,2]*, Sonja Meimann[2], Irina Solovei[3], Irene L. Gügel[4], Christina Geiger[5], Nicole Schauerte[5], Karolina Goździewska-Harłajczuk[6], Joanna E. Klećkowska-Nawrot[6], Gudrun Wibbelt[7], Silke Haverkamp[4]

**1** Institute for Clinical Neuroanatomy, Dr. Senckenbergische Anatomie, Goethe University, Frankfurt am Main, Germany, **2** Institute of Cellular and Molecular Anatomy, Dr. Senckenbergische Anatomie, Goethe University, Frankfurt am Main, Germany, **3** Biozentrum, Ludwig Maximilians University Munich, Planegg-Martinsried, Germany, **4** Department of Computational Neuroethology, Max Planck Institute for Neurobiology of Behavior-Caesar, Bonn, Germany, **5** Veterinary Department, Zoo Frankfurt, Frankfurt am Main, Germany, **6** Department of Biostructure and Animal Physiology, Faculty of Veterinary Medicine, Wrocław University of Environmental and Life Sciences, Wrocław, Poland, **7** Department of Wildlife Diseases, Leibniz Institute for Zoo and Wildlife Research, Berlin, Germany

* peichl@em.uni-frankfurt.de

## Abstract

The nocturnal aardvark *Orycteropus afer* is the only extant species in the mammalian order Tubulidentata. Previous studies have claimed that it has an all-rod retina. In the retina of one aardvark, we found rod densities ranging from 124,000/mm² in peripheral retina to 214,000/mm² in central retina; the retina of another aardvark had 163,000 – 245,000 rods/mm². This is moderate in comparison to other nocturnal mammals. With opsin immunolabelling we found that the aardvark also has a small population of cone photoreceptors. Cone densities ranged from about 300 to 1,300/mm² in one animal, and from 1,100 to 1,600/mm² in a limited sample of the other animal, with a central-peripheral density gradient and some local variations. Overall, cones comprised 0.25-0.9% of the photoreceptors. Both typical mammalian cone opsins, longwave-sensitive (L) and shortwave-sensitive (S), were present. However, there was colocalization of the two opsins in many cones across the retina (35 – 96% dual pigment cones). Pure L cones and S cones formed smaller populations. This probably results in poor colour discrimination. Thyroid hormones, important regulators of cone opsin expression, showed normal blood serum levels. The relatively low rod density and hence a relatively thin retina may be related to the fact that the aardvark retina is avascular and its oxygen and nutrient supply have to come from the choriocapillaris by diffusion. In contrast to some previous studies, we found that the aardvark eye has a reflective tapetum lucidum with features of a choroidal tapetum fibrosum, in front of which the retinal pigment epithelium is unpigmented. The discussion considers these findings from a comparative perspective.

**Data availability statement:** All relevant data are within the manuscript and its Supporting Information files.

**Funding:** I.S. received grants SP2202/SO1054/2 project #422388934, and SFB1064 project #213249687 from the Deutsche Forschungsgemeinschaft (https://www.dfg.de/en) The funder had no role in study design, data collection and analysis, decision to publish, or preparation of the manuscript.

**Competing interests:** The authors have declared that no competing interests exist.

## Introduction

The aardvark (*Orycteropus afer*) is the only extant species of the order Tubulidentata in the supraordinal mammalian clade Afrotheria. It represents a very early branch on the Afrotheria phylogenetic tree (Fig 1) [1,2]. The aardvark is a medium-sized, pig-like mammal native to sub-Saharan Africa (Fig 2). It is a nocturnal, burrowing species that mostly feeds on ants and termites (for an overview, see, e.g., [3,4]). The strong legs with sharp claws are used to dig out termite and ant nests, as well as abode burrows; the genus name *Orycteropus* means 'burrowing foot.' The aardvark is considered a 'living fossil,' because *Orycteropus* fossils from about 20 million years ago show nearly identical morphological features to those of living aardvarks [4]. It is assumed that aardvarks have an acute sense of smell and hearing, but poor eyesight. However, in contrast to the abundant literature available on the eyes and retinae of many other mammals, the only substantial study of the aardvark retina known to us is that of Victor Franz, published in 1909 [5]. Franz obtained the two eyes of one animal hunted at a zoological expedition and examined them in detail macroscopically and microscopically. The study contains much valuable information, but also some apparent errors, most likely

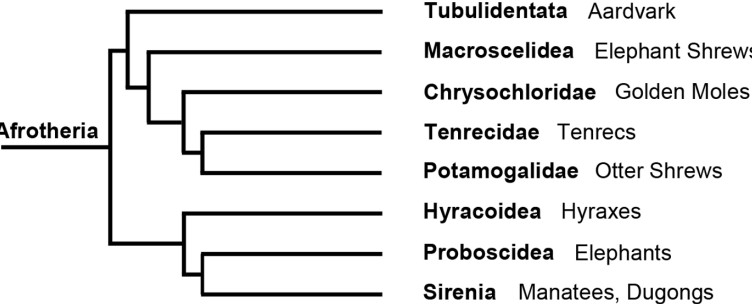

**Fig 1. Evolutionary tree of the Afrotheria, showing the isolated position of the Tubulidentata with the aardvark as its only extant species.** The tree is a schematic representation based on [1], the timeline of branch points is not to scale.

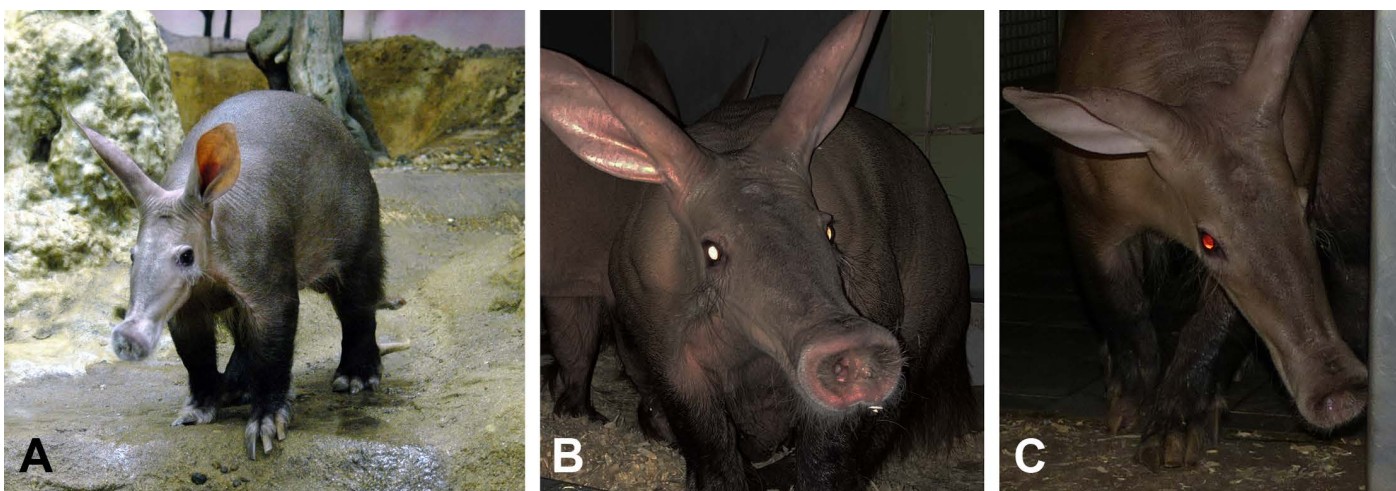

**Fig 2. Aardvarks at day and night, note the laterally positioned eyes.** (A) Aardvark 1, the post mortem donor of the studied eye, at daytime. (B, C) Another aardvark, flashlight photographs taken at night. When the head is viewed horizontally, there is a bright whitish eyeshine indicating a tapetum lucidum; when viewed from above, the weaker eyeshine appears orange to red. For details see text. Image sources: (A) Frankfurt Zoo; (B, C) Christina Geiger.

due to the histological methods available at the time. Franz [5] reports a complete absence of cone photoreceptors in the retina, which appears doubtful in the light of more recent research demonstrating cones in nearly all mammals studied to date (reviews: [6,7]). Among the few exceptions without functional cones are some deep-diving whales [8], the subterranean golden moles [9], and Xenarthra [10]. Franz [5] also states that the aardvark has no reflective tapetum lucidum. In contrast, a recent study of the orbital structures and eye tunics of young and adult aardvarks reports the presence of a tapetum lucidum [11]. Nocturnal photographs of aardvarks show a strong eyeshine or 'glow' (Fig 2). However, this could also be a reflection off the fundus like the 'red eye effect' seen, e.g., in human eyes in flash photographs. When we obtained the relatively well-preserved eyes of two aardvark individuals, we studied them with currently available histological approaches to resolve the above discrepancies and to add new observations on aardvark retinal anatomy. Here we report on general eye features and the photoreceptors.

## Materials and methods

### Tissue

This study has used tissue from aardvarks (*Orycteropus afer*) that was obtained from zoo animals that had died of natural causes. The aardvark IUCN status is 'least concern' and no ethical approval was required for use of the tissue. The right eye of a nearly 25 years old male aardvark was obtained when the animal died of old age in the Zoo of Frankfurt am Main, Germany. The animal is termed "aardvark 1" here (Fig 2A). It had an age-related cataract and a mild chronic *Uveitis anterior*, but the retina appeared macroscopically normal. One day post mortem, the eye was enucleated during autopsy, punctured behind the cornea for better fixative penetration, and immersion-fixed in 4% formalin for 24 h at 4°C. When punctured, the eye lost some liquefied vitreous and aqueous humor, leading to a collapse of the cornea and a slight deformation of the eyeball during fixation. After fixation the external eye dimensions were recorded, the eye was cut open behind the cornea, and the posterior eyecup with attached retina was washed in 0.01M phosphate buffered saline (PBS, pH 7.4). The retina was carefully dissected from the eyecup after having marked its orientation. It was cryoprotected by successive immersion in 10%, 20% and 30% (w/v) sucrose in phosphate buffer (PB, pH 7.4) containing 0.05% sodium azide, and stored frozen at -20°C until further processing. The eyecup was stored in PBS with 0.05% sodium azide at 4°C.

The eyes of a nearly 6 years old female aardvark were obtained when the animal died of perinatal complications in the Zoological Garden of Wrocław (Poland). The animal is termed "aardvark 2" here, it was genetically unrelated to aardvark 1. The eyes were enucleated immediately post mortem and immersion-fixed in 4% buffered formaldehyde solution, they have been used in a previous study of aardvark eye and orbital features [11]. The right eye was kept in the fixative for 1 week and then embedded in paraffin for sectioning. The left eye was permanently stored in the fixative, and only small pieces of peripheral retina were available for the present study. Probably due to the long fixation time of this eye (approx. 7 years), some of the antibodies used here did not work (see below and Results).

For frozen vertical sections of the retina (i.e., perpendicular to the retinal layers), pieces of retina were transferred from 30% sucrose to tissue freezing medium (Leica Biosystems, Wetzlar, Germany), frozen, sectioned at 16 μm thickness with a cryostat (Leica CM 3050 S, Wetzlar, Germany), and collected on Superfrost Plus slides (Menzel Gläser, Braunschweig, Germany).

For electron microscopy, small pieces of the sclera and presumed tapetum lucidum were stained as previously described [12]. Briefly, the samples were stained in a solution containing

1% osmium tetroxide, 1.5% potassium ferrocyanide, and 0.15 M cacodylate buffer. The osmium stain was amplified with 1% thiocarbohydrazide and 2% osmium tetroxide. The tissue was then stained with 2% aqueous uranyl acetate and lead aspartate. The tissue was dehydrated through an 70%–100% ethanol series, transferred to propylene oxide, infiltrated with 50%/50% propylene oxide/epon medium hard formulation (EMbed 812, Electron Microscopy Sciences; [13]), and then 100% epon medium hard. The epon medium hard infiltrated tissue was transferred into multi-well embedding molds (Electron Microscopy Sciences) and hardened at 60°C. For scanning electron microscopy (SEM), a few serial sections of 50 nm were taken with a Diatome ultra diamond knife and collected on glow discharged silicon wafers and dried on a heating plate at 50 °C until the water was fully evaporated. The wafers were mounted with silver paint (Plano) on a sample holder and images were taken with a Supra55 (Leica) SEM. For transmission electron microscopy (TEM), 50-nm-thick sections were cut with an Ultra diamond knife and transferred on carbon-coated copper grids with a hole size of 35/10 nm (S35/10, Quantifoil, Electron Microscopy Sciences). Images were recorded with an analytical electron microscope (JEM-2200FS, Jeol) at an energy of 200 keV with a CMOS camera (TEM-CAM F416, TVIPS).

For comparison of the choroid, vertical cryo-sections of the formalin-fixed eye of a captive adult African elephant (*Loxodonta africana*) from a German zoo was used. In agreement with CITES regulations the eye was collected during necropsy for pathological investigations performed by the Institute for Zoo and Wildlife Research, Berlin (IZW) after the animal had to be euthanized because of severe disease unresponsive to treatment.

## Immunohistochemistry

Immunohistochemistry was performed on vertical sections of the retina, as well as on unsectioned retinal pieces from various regions to assess cell populations in flat view. Immuno-labelling followed standard protocols. Briefly, sections on the slide and free-floating pieces were preincubated for 1 h in PB with 0.5% Triton X-100 and 10% normal donkey serum (NDS). Incubation in the primary antibody/antiserum solution, made up in PB with 3% NDS and 0.5% Triton X-100, was overnight at room temperature for sections, and 3 days at room temperature or 4 days at 4°C for unsectioned pieces. Multiple immunofluorescence labelling for simultaneous visualization of several antigens was performed by incubation in a mixture of the antisera. Table 1 lists all primary antibodies used. The cone opsin antisera JH492, JH455 and sc-14363 have been used in several previous studies to reliably label the respective opsins in a range of mammals [16–20].

Binding sites of the primary antibodies were visualized by indirect immunofluorescence, with a 1.0-1.5 h incubation of the tissue in the secondary antiserum, or in a mixture of appropriate secondary antisera in the case of several primary antibodies. We used secondary antisera conjugated to Alexa 488, Alexa 647, Cy3 and Cy5 in appropriate combinations. Omission of the primary antibodies from the incubation solution resulted in no staining. In addition to antisera, we used the fluorescent markers peanut agglutinin (PNA) and NeuroTrace for certain cell types (Table 1). After immunolabelling, sections were incubated in a solution of 4,6-diamidino-2-phenylindole (DAPI) as a fluorescent nuclear stain to reveal the general retinal layering. Choroidal blood vessels were labeled by biotinylated isolectin B4 (Table 1; incubation was 1 h for sections, 2 h for unsectioned pieces) and visualized by a subsequent 1.0-1.5 h incubation in streptavidin coupled to Alexa 549 or 488. Tissue was coverslipped with an aqueous mounting medium (AquaPoly/Mount, Polysciences Inc., Warrington, PA, USA; or Dako Fluorescence Mounting Medium S3032, Dako North America Inc., Carpinteria, CA, USA).

In retinal pieces from the long-fixed eye of aardvark 2, the S cone opsin antiserum sc14363 did not work, hence assessment of the cone opsin pattern was done by sequential

double-labeling with the two rabbit antisera JH492 against the L cone opsin and JH455 against the S cone opsin as follows. One retinal piece was first incubated in a JH492 solution and then in a donkey-anti-rabbit antiserum conjugated to Alexa 488. Then the piece was incubated in a JH455 solution and finally in a donkey-anti-rabbit antiserum conjugated to Cy3. This secondary antiserum bound to both primary antisera, hence all cones were labeled by Cy3. The cones also labeled by Alexa 488 were those that contained L cone opsin, and cones only labeled by Cy3 were pure S cones. In a neighboring piece of retina, the order of labeling was reversed: First incubation in the JH455 solution and visualization with the donkey-anti-rabbit antiserum conjugated to Alexa 488, then incubation in the JH492 solution and visualization with the donkey-anti-rabbit antiserum conjugated to Cy3. Again, all cones were labeled by Cy3, but here the cones also labeled by Alexa 488 were those that contained S cone opsin, and those only labeled by Cy3 were pure L cones. Combining the data from the two pieces provided the

**Table 1. Primary antibodies and cell markers used.**

| Antigen/marker | Immunogen/ target structure | Antibody host species, catalog #, RRID | Dilution | Source |
|---|---|---|---|---|
| Rod opsin RH1 | N-terminal region of bovine rhodopsin RH1[1] | Mouse monoclonal, Name: rho4D2, RRID: AB_2315273 | 1:1,000 | Gift of R. S. Molday, University of British Columbia Life Sciences Centre, Vancouver, Canada |
| L cone opsin LWS | C-terminal 38 amino acids of human red cone opsin[2] | Rabbit polyclonal, Name: JH 492, RRID: AB_2315259 | 1:2,000 | Gift of J. Nathans, Johns Hopkins University School of Medicine, Baltimore, Maryland, USA |
| S cone opsin SWS1 | C-terminal 42 aa of human blue cone opsin[2] | Rabbit polyclonal, Name: JH 455, RRID: AB_2313807 | 1:5,000 | Gift of J. Nathans, Johns Hopkins University School of Medicine, Baltimore, Maryland, USA |
| S cone opsin SWS1 | 20 aa peptide mapping near N-terminus of human blue cone opsin[3] | Goat polyclonal, Cat# sc-14363, RRID: AB_2158332 | 1:500 | Santa Cruz Biotechnology |
| CtBP2 (C-terminal Binding Protein-2) | Mouse CtBP2, aa. 361-445 | Mouse monoclonal, Cat# 612044, RRID: AB_399431 | 1:5,000 | BD Biosciences |
| CtBP2 | Rat CtBP2, aa 431-445 | Rabbit polyclonal, Cat# 193 003, RRID: AB_2086768 | 1:5,000 | Synaptic Systems |
| Glutamine synthetase | Müller glia | Mouse monoclonal, Cat# 610517, RRID: AB_397879 | 1:500 | BD Bioscience |
| GFAP | Glial fibrillary acidic protein | Mouse monoclonal, Cat# G3893, RRID: AB_477010 | 1:500 | Sigma-Aldrich |
| H3K4me3 | Euchromatin | Rabbit polyclonal, Cat# ab8580, RRID: AB_306649 | 1:500 | abcam |
| H4K20me3 | Heterochromatin | Mouse monoclonal, CMA423 | 1:500 | Generated in Hiroshi Kimura' lab, Tokyo University |
| Lamins A/C | Inner nuclear membrane protein | Mouse serum | Undiluted | Gift of Harald Herrmann, German Cancer Research Center |
| LBR | Inner nuclear membrane protein | Guinea pig serum | 1:50 | Generated in Harald Herrmann's lab, German Cancer Research Center |
| PNA-647 (Peanut agglutinin) | General cone marker | Cat# L-32460 | 1:100 | Molecular Probes |
| NeuN | Synthetic peptide NeuN | Rabbit monoclonal (EPR12763), Cat# ab177487, RRID: AB_2532109 | 1:100 | abcam |
| NeuroTrace (Ex 530/ Em 615) | Fluorescent Nissl stain | Cat# N-21482 | 1:100 | Molecular Probes |
| Isolectin B4, biotinylated | Blood vessel marker | Cat# B-1205 | 1:50 | Vector Laboratories |

[1]Ref. [14],

[2]Ref. [15],

[3]Ref. [16].

total cone density, the percentages of pure L and S cones, and the proportion of cones containing L and S opsin, from which the percentage of dual pigment cones could be calculated for that retinal region.

For assessment of the retinal vascularization, a retinal piece of 5.5 x 4.0 mm size containing the optic nerve head was stained with a 3,3'-diaminobenzidine (DAB) reaction to selectively visualize the endogenous peroxidase in the vasculature. The retinal piece was washed in 0.05% Tris buffer (TRIS, pH 7.6), then incubated for 20 min in a solution of 0.05% DAB in TRIS, after which hydrogen peroxide was added to the incubation solution at a final concentration of 0.01%, and the incubation was continued for 10 min until the peroxidase reaction had fully developed. The reaction was stopped by several washes in TRIS and then PB. The retinal piece was flat-mounted on a slide and coverslipped with AquaPoly/Mount. For assessment of the retinal pigment epithelium (RPE) after removal of the retina, pieces of the thin RPE layer from the central and peripheral fundus were gently removed from the underlying tissue, flat-mounted on a slide and coverslipped with AquaPoly/Mount.

### Imaging and analysis

The RPE and the DAB-labelled vasculature of the optic nerve head were analyzed with a Zeiss Axioplan 2 microscope by differential interference contrast. Micrographs were taken with a CCD camera and the Axiovision LE software (Carl Zeiss Vision, Germany). The immunofluorescence-labelled sections and retinal pieces were analyzed with a laser scanning microscope (LSM) Olympus FluoView 1000 using the FV 1.7 software (Olympus), or with a Leica TCS SP5 or a Leica TCS SP8 confocal microscope. LSM images and z-stack projections were examined with ImageJ (https://imagej.net); cells were counted using the cell counter plugin. Images for illustration were adjusted for brightness and contrast using Adobe Photoshop. Irrespective of the fluorescent dye used to visualize a label, labels are shown in the RGB channels that are most suitable to illustrate label combinations. For the benefit of red/green-blind readers, combinations of magenta and green are preferred over red and green.

### Thyroid hormone

Thyroid hormone (TH), via its receptor TRβ2, is an important regulator of cone spectral identity by repressing S opsin and activating L opsin in developing and adult retina (see Discussion). Because of the L and S opsin co-expression in a large proportion of the aardvark cones, we were interested to know whether the serum TH levels in aardvark differ from those in other mammals. Frankfurt Zoo, during medical check-ups, had obtained five blood counts of aardvark 1 over the last three years of his life, and one blood count of his 21 years old son (here termed "aardvark 3"), all including serum TH levels. The blood analysis was done by the commercial veterinary clinical diagnostics laboratory LABOKLIN (Bad Kissingen, Germany).

## Results

Most findings reported here came from aardvark 1. The retina of aardvark 2 was used for comparison of the photoreceptor findings.

### General eye features

The eye of aardvark 1 had an equatorial diameter of approx. 23.0 mm and an axial length of approx. 21.2 mm. The axial length probably is an underestimate because the cornea had collapsed (Fig 3A) and its original curvature could only be estimated. The cornea was elliptical with a naso-temporal diameter of 18.8 mm and a dorso-ventral diameter of 15.5 mm (mean 17.1 mm); the ratio of mean corneal diameter to eye equatorial diameter was 0.75, and the

ratio of mean corneal diameter to eye axial length was 0.81. The lens diameter was 13.3 mm and the lens thickness 9.5 mm (Fig 3B). The curvature was stronger at the posterior than at the anterior side of the lens (not illustrated). The ratio of lens diameter to eye equatorial diameter was 0.58, the ratio of lens thickness to eye axial length was 0.45. The poor pigmentation of the choroid and sclera in the present eye confirms the observations by Franz [5].

The fundus in the opened eyecup showed a bright yellowish band extending from the temporal to the nasal periphery. This bright band had the appearance of a tapetum lucidum, its dorso-ventral width was larger in the temporal than the nasal fundus. Its relatively sharp ventral boundary ran along the horizontal midline of the fundus, its dorsal boundary showed a more gradual transition to the pigmented part of the fundus (Fig 3C and D). The ventral half and the dorsal periphery of the fundus were covered by brown-black retinal pigment epithelium (RPE). The optic nerve head (optic disc, OD) was located centrally on the temporal-nasal eye axis and ventral to the geometric center of the eyecup, about one OD diameter below the ventral boundary of the bright fundus band (Fig 3C and D). When the retina was removed, the remaining thin RPE layer consisted of RPE cells (melanocytes) that contained a high density of melanin granules in the dark-appearing parts of the fundus (Figs

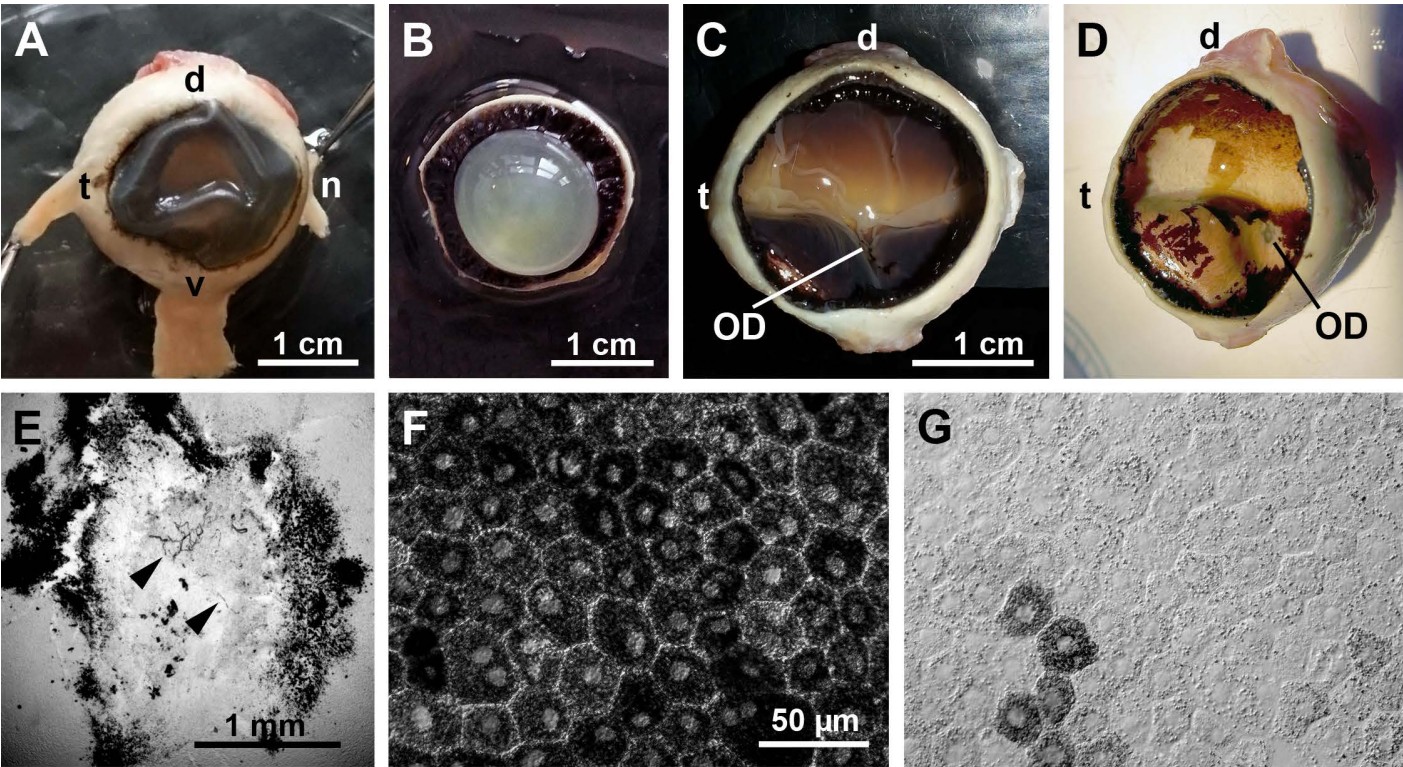

**Fig 3. General eye features (aardvark 1).** (A) Intact eye, frontal view with attached rectus muscles. The cornea had collapsed when the eye was punctured for better fixative penetration to the retina. (B) Anterior part of the opened eye from the vitreous side, showing the lens and ciliary body. (C) Opened eyecup showing the fundus with the retina in situ. There is a bright horizontal band where the retinal pigment epithelium (RPE) is weakly pigmented or unpigmented. The transition to the pigmented peripheral RPE is gradual at the dorsal side and with a rather sharp boundary at the ventral side. The optic disc (OD) is located ventral to the bright horizontal band. (D) Eyecup after removal of the retina. Some RPE also came off during the preparation, showing an unpigmented, whitish-yellow choroid. Here, the optic disc (OD) is more obvious than in (C). (E) Optic disc in the isolated retina, DAB-reacted for blood vessels. There are only a few capillaries present in the OD (arrow heads), and no blood vessels exit it to supply the surrounding retina. Around the OD there is an accumulation of pigment. (F, G) Light microscopic images of flat-mounted RPE pieces from peripheral (F) and central fundus (G). In the periphery, all RPE cells contain densely packed melanin granules (F); centrally, only very few RPE cells are rich in melanin granules, the vast majority of RPE cells contains little or no melanin (G). Eye dimensions in (D) can be determined from the scale bar in (C). The scale bar in (F) applies to (F, G). d, dorsal; n, nasal; t, temporal; v, ventral.

3D,F, 4A). In the central bright-appearing band, the large majority of RPE cells contained very few or no melanin granules, and only some single cells or small cell clusters contained ample melanin (Figs 3G and 4A). The absence of pigmentation in a horizontal band of the central fundus and the associated tapetum-like reflection explain the eyeshine differences seen at different angles (Fig 2B and C). When the eye is seen horizontally, the strong reflectivity of this band produces a bright whitish to yellowish eyeshine. When the eye is seen from above, the lower reflectivity of the more pigmented ventral fundus produces a fainter orange to red eyeshine that resembles the 'red eye effect' seen in flash photographs of human eyes with their pigmented fundus.

## Choroid and tapetum lucidum

Once the retina was removed, the thin RPE layer readily detached from the choroid in shreds (Figs 3D and 4A). The strongly vascularized choroid was unpigmented and appeared whitish with a mother-of-pearl-like reflection throughout the fundus; in the fundus regions with pigmented RPE, this choroidal reflection was concealed (Fig 4A). Electron microscopy of transverse choroid sections showed that at the RPE-facing side of the choroid, there were fibrillar structures arranged in parallel, with different orientations in neighbouring domains (Fig 4B). In some domains the fibrils were densely packed, in others they were less dense or sparse. Fibril diameters seen in cross-sections were 120-310 nm, being 150-250 nm in most cases. Observed fibril lengths were up to about 7 μm. In TEM images, the fibrils showed the typical cross-striation of native collagen, indicating a choroidal tapetum lucidum fibrosum (Fig 4C). However, in SEM images, cross-sections of the fibrils showed a substructure with a more electron-dense shell and core, which is more characteristic of the rodlets of a tapetum cellulosum (Fig 4D), see Discussion. Overall, the presumed tapetum layer is thinner and less conspicuously striated than, e.g., in the African elephant which, like the aardvark, belongs to the Afrotheria and has an avascular retina (Fig 4E and F). The boundary of the choroid and tapetum to the RPE is formed by the choriocapillaris, a dense capillary net that was labelled by the endothelial cell marker isolectin B4 in vertical sections of both aardvark (Fig 4G) and African elephant (Fig 4H). The dense mesh of the aardvark choriocapillaris is particularly obvious in flat view (Fig 4I).

## General Retina Features

We confirm that the aardvark retina is avascular. In the fundus, there were no obvious blood vessels emerging from the optic disc or extending across the retina. Staining of blood vessels with DAB in a piece of central retina containing the optic disc revealed a few small capillaries within the optic disc and confirmed that there were no blood vessels extending outside the optic disc and into the retina (Fig 3E).

In the vertical sections, retinal thickness ranged from approx. 120 μm to 180 μm. This is an estimate, given the fact that the sections may not be exactly vertical and that the length of photoreceptor outer segments may not be fully preserved. The layering of the aardvark retina conformed to the typical mammalian pattern (Fig 5). The outer nuclear layer (ONL) with the photoreceptor somata was the thickest layer, indicating a dominance of rod photoreceptors, which is the situation seen in most mammals. The ONL had six to nine soma tiers in more central retina and five to seven tiers in more peripheral retina. The inner nuclear layer (INL) had three to four soma tiers in central retina and two to three soma tiers in peripheral retina. The ganglion cell layer (GCL) was sparsely populated by somata. The narrower outer plexiform layer (OPL) and broader inner plexiform layer (IPL) separated the soma layers.

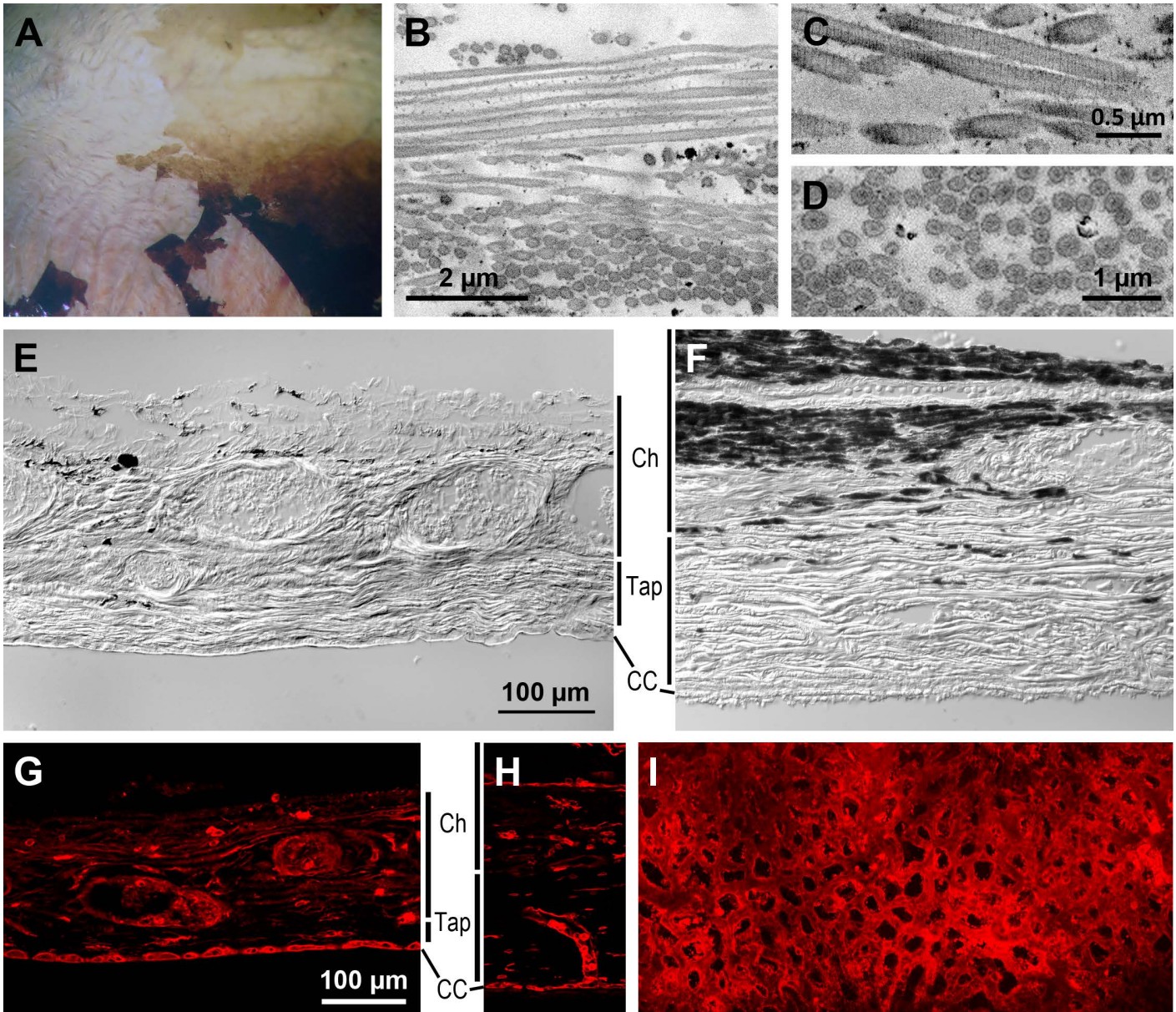

**Fig 4. Choroid and tapetum lucidum (aardvark 1).** (A) Higher power view of the central and ventral midperipheral fundus of the aardvark eye (c.f. Fig 3C and D). Below the partly removed RPE layer, the whitish unpigmented choroid with its red blood vessels is visible. (B) SEM micrograph of subcellular structures from the RPE-facing side of a transverse choroid section, showing collagen fibrils arranged in parallel bundles with different orientations that indicate a tapetum lucidum. In the upper image part, the fibrils are longitudinally sectioned; in the bottom image part, they are cross-sectioned with round to oval profiles. (C) At higher magnification, the fibrils show the typical cross-striation of native collagen (TEM image). (D) In cross-section, the fibrils show a shell and core of higher electron density (SEM image). (E) Differential interference contrast (DIC) image of a vertical cryo-section of the aardvark choroid (Ch) and tapetum (Tap). The poorly pigmented choroid shows cross-sections of blood vessels of various calibers, the tapetum shows a horizontal striation indicating tapetal laminae. (F) DIC image of a vertical cryo-section of the choroid and tapetum of an African elephant for comparison. The choroid is strongly pigmented, and the tapetum is thicker and more conspicuously striated than in the aardvark. (G) Aardvark vertical cryo-section of the choroid and tapetum with blood vessel labelling by isolectin (red). The choroid part contains vessels of larger and smaller caliber, the tapetum layer is relatively thin, and the choriocapillaris (CC) at the border to the RPE is densely filled with capillaries. (H) African elephant vertical cryo-section of the choroid and tapetum with blood vessel labelling by isolectin (red) for comparison. The image shows a vertical choroidal blood vessel supplying the CC capillaries. For the sections of (E-H), the choroid has been removed from the sclera, so the sections do not show the full thickness of the choroid. (I) Flat view of the aardvark choriocapillaris, labelled by isolectin (red) and showing the dense capillary net. The scale bar in (E) applies to (E, F), the scale bar in (G) applies to (G-I).

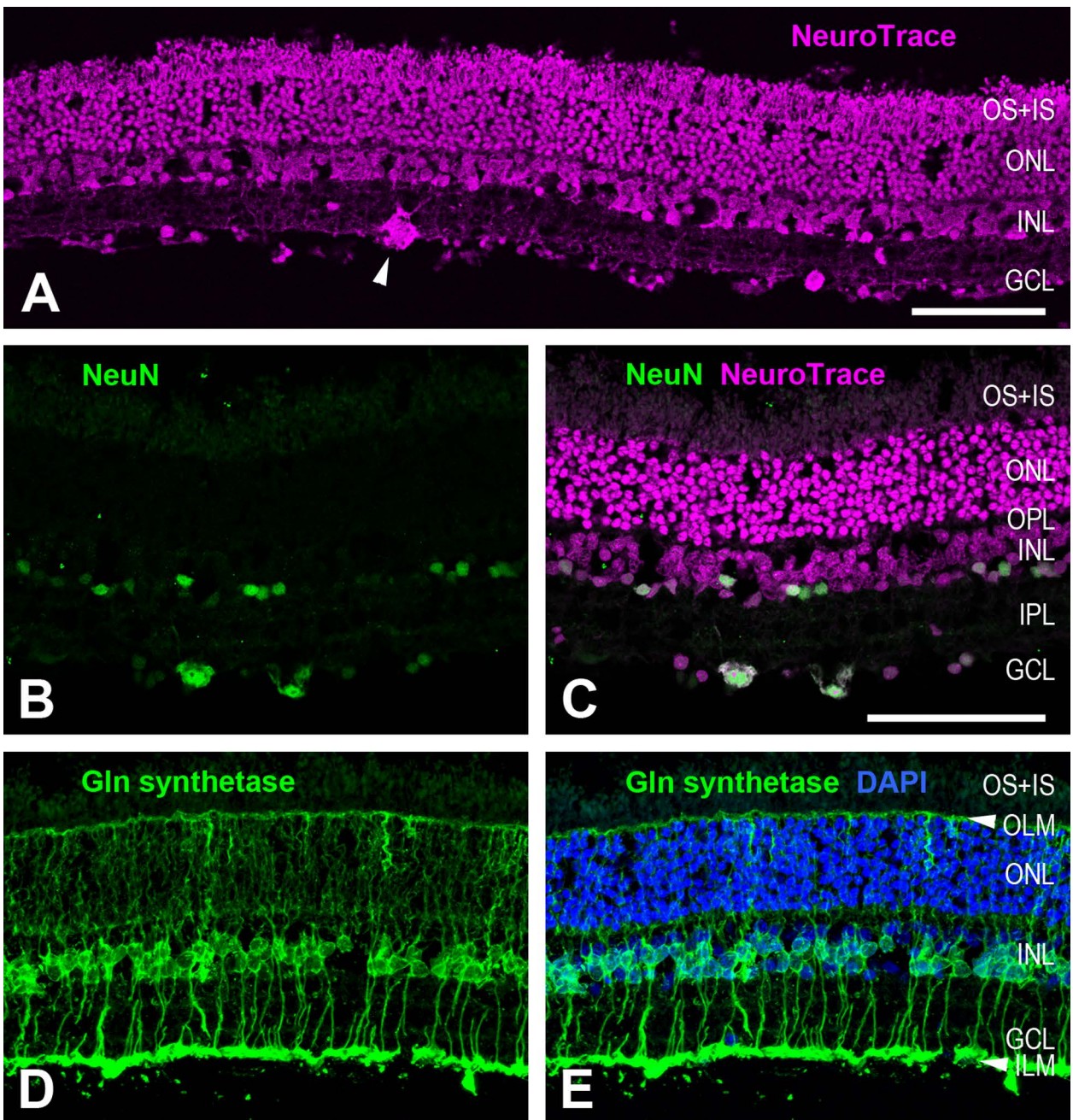

**Fig 5. General retinal features (aardvark 1).** (A) Overview of a vertical retinal section labelled with the fluorescent Nissl stain NeuroTrace, revealing the retinal layering. Cell bodies in the outer nuclear layer (ONL) and inner nuclear layer (INL) are stacked in several tiers. The photo-receptor outer and inner segments (OS+IS) are also labelled. The ganglion cell layer (GCL) is sparsely populated by cells of various soma sizes. A large soma of a putative alpha ganglion cell is marked by an arrowhead. (B, C) Double labelling with an antibody against the neuronal marker NeuN and with NeuroTrace. NeuN only labels a few presumed amacrine cell somata in the INL and some somata in the GCL (B). The Neu-roTrace counterstain shows the position of the NeuN somata in the layers (C). (D, E) Immunolabelling for glutamine synthetase shows the Müller cells forming the retinal glia scaffold (D). Counterstaining with DAPI (E) shows that the Müller cells have their somata in the INL and vertically extend their processes from the inner limiting membrane (ILM) formed by their endfeet to the outer limiting membrane (OLM). The images are maximum intensity projections of confocal image stacks. OPL, outer plexiform layer; IPL, inner plexiform layer. Scale bars are 100 μm, scale bar in (C) applies to (B-E).

Müller cells, the scaffolding radial macroglia of the retina, were specifically labelled by an antibody against glutamine synthetase (Fig 5D and E). They were present at a high density and had the mammalian-typical morphology. Their somata were located in the INL. Their inward processes traversed the IPL and terminated in the Müller cell endfeet that ended at the inner limiting membrane (ILM), separating the retina from the vitreous. In some of the cells these processes bifurcated and formed two endfeet. Their outward processes encircled the photoreceptor somata in the ONL and ended at the outer limiting membrane (OLM) that lies between the ONL and the photoreceptor inner segments. An antibody against glial fibrillary acidic protein (GFAP) did not label any structures (not illustrated). Hence, there are no astrocytes in the aardvark retina, as mammalian astrocytes specifically express GFAP [21]. This is in line with the absence of retinal blood vessels (see Discussion). Furthermore, the absence of GFAP label in the Müller cells indicates that the studied retina was healthy. The Müller cells of healthy retinae have very low or no GFAP expression, but show reactive gliosis with dramatically upregulated GFAP expression to practically all forms of retinal stress, i.e., to various retinal diseases and injuries (reviews: [22,23]).

## Rod photoreceptors

In the retina of aardvark 1, rod photoreceptors were identified by labelling with an antibody to the mammalian rod opsin RH1 (Fig 6A–C), and by their characteristic nuclear morphology (Fig 6E). Their outer segments showed the most intense RH1 label and formed a densely packed layer at the outer retinal surface (Fig 6A). The used antibody rho4D2 also, but less intensely, labelled the other parts of the rod cells, particularly the soma cytoplasm in the ONL and the axonal ending in the OPL (Fig 6B and C), which is common in mammals. The vast majority of the photoreceptor somata in the ONL showed rho4D2 label and hence are rod somata; only the few cone somata that were identified by their different nuclear morphology (Fig 6C and F) lacked cytoplasmic rho4D2 label (Fig 6C). This fits the very low aardvark cone densities (see below).

In the vast majority of eukaryotic cells, the euchromatin is located in the centre of the nucleus and the heterochromatin in the nuclear periphery. In contrast, the rod nuclei of nocturnal mammals have a unique inverted chromatin architecture with heterochromatin aggregated in the nuclear centre and euchromatin arranged at the nuclear periphery. This unusual chromatin arrangement evolved as an adaptation to night vision because it reduces light scattering in the thick retinae of nocturnal species, enhancing their ability to detect low intensity light [24–26]. In the nocturnal aardvark, the heterochromatin in the rods was clustered in two large central granules, thus the nuclei seem to be inverted (Fig 6E,I,J). At the same time, the internal heterochromatin exhibited several protrusions towards the nuclear envelope (Fig 6E), making these nuclei semi-inverted. All other retinal cells, including the cones, had the conventional nuclear architecture with heterochromatin attached to the nuclear periphery or nucleolus (Fig 6F–H). The nuclear envelope protein lamin A/C was present in all retinal nuclei including the rod nuclei, whereas the lamin B receptor (LBR) of the nuclear envelope was only present in retinal microglial cells, not in any retinal neurons (Fig 6K–M). The functional implications of the nuclear inversion and the role of nuclear envelope proteins in this process are addressed in the Discussion.

In the retina of aardvark 1, photoreceptor densities (and hence rod densities) were estimated from counts of ONL nuclei in DAPI-stained vertical sections at 16 positions from the visual streak region, midperipheral and peripheral retina. The sections ran from the OD to the dorsal edge of the retina (S1 Fig). The observed range was about 124,000 – 214,000 photoreceptors (rods)/mm² , with densities basically decreasing from central to peripheral retina, but also showing marked local variations (S1 Table). Concomitantly, the ONL thickness decreased

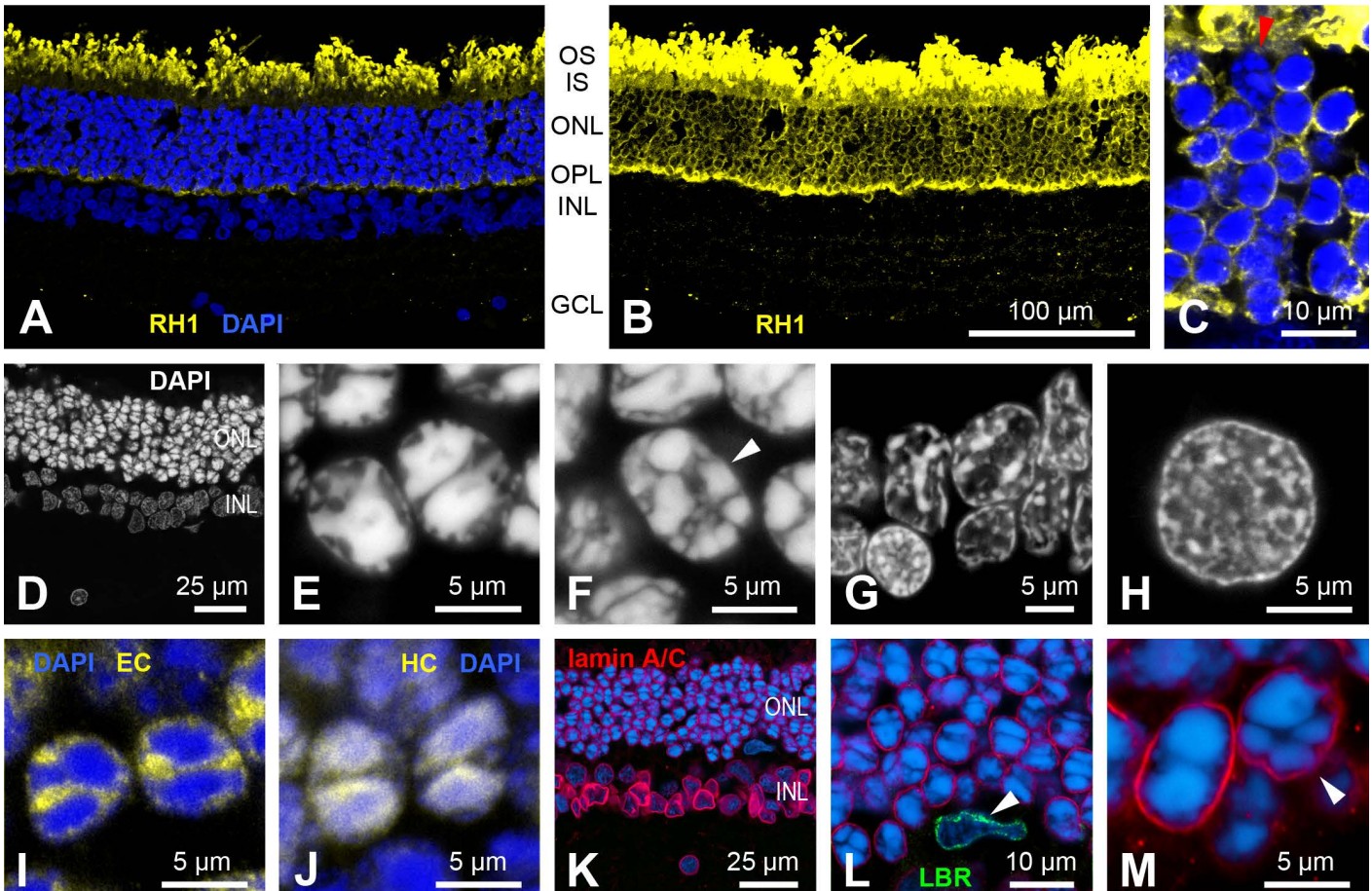

**Fig 6. Rod photoreceptors (aardvark 1).** (A, B) Vertical retinal section, rod opsin RH1 label (yellow). (A) The rod outer segments are strongly labelled, DAPI counterstaining (blue) shows the retinal nuclear layers. (B) Overexposure of the same field shows the less strong RH1 label in the rod somata in the outer nuclear layer (ONL) and the rod axonal spherules in the outer plexiform layer (OPL). Clearly, the vast majority of ONL somata belong to rods. (C) High-power view of the ONL showing one RH1-negative cone soma (red arrowhead) among RH1-labelled rod somata. (D-H) DAPI nuclear staining of various retinal neurons to reveal their heterochromatin arrangement. (D) Overview of a DAPI stained section, nuclei in the ONL are more intensely stained than those in the INL, confirming the appearance seen in (A). (E) The rod nuclei show a "semi-inverted" nuclear architecture with most of the heterochromatin clustered in the nuclear centre, often in two aggregates, but with some extensions towards the nuclear periphery. (F) In contrast, the cone nuclei (arrowhead) have several smaller heterochromatin clusters localized towards the nuclear periphery (conventional nuclear architecture). The same conventional heterochromatin arrangement is seen in cells of the INL (G) and in retinal ganglion cells (H). (I) In the rods, euchromatin (immunolabelled by anti-H3K4me3, yellow) is located mostly in the nuclear periphery and in the gaps between the heterochromatin clusters (DAPI, blue). (J) In the rod nuclei, heterochromatin (immunolabelled by anti-H4K20me3, yellow) colocalizes with the DAPI staining (blue), the merge of the labels appears whitish. (K-M) Immunolabelling for lamin A/C (red) and LBR (green), counterstained with DAPI (blue). The aardvark retina shows a presence of lamin A/C in the neuronal nuclei in all layers (K, L). As a positive control for labeling with the anti-LBR antibody, the nucleus of a microglial cell (arrowhead) expressing LBR but not lamin A/C is shown in (L). LBR label is only present in microglial cells, not in any neurons. (M) A rod nucleus (left) and a cone nucleus (right, arrowhead) with labelled lamin A/C. For layer abbreviations, see Fig 5. Scale bar in (B) applies to (A, B).

from eight to nine tiers of photoreceptor somata in central to six tiers in peripheral retina; examples from central and peripheral retina are shown in Fig 6A and B, and in Fig 6D and K, respectively. In the retina of aardvark 2, photoreceptor densities were estimated from counts of ONL somata at seven positions in vertical paraffin sections of one eye from central (near OD), midperipheral, and peripheral retina. Here the range was about 163,000 – 245,000 photoreceptors (rods)/mm² (S1 Table). The ONL had five to nine tiers of photoreceptor somata. It is possible that aardvark 2 had higher rod densities than aardvark 1, but the larger shrinkage of paraffin-embedded tissue may also have artefactually increased the cell density.

## Cone photoreceptors

The cone photoreceptors were identified by labelling with antisera to the longwave-sensitive (L) cone opsin and the shortwave-sensitive (S) cone opsin in retinal sections (Fig 6A and B) and in flat-mounted retinal pieces from various retinal regions (Fig 7C and S1 Fig). The two S opsin antisera sc14363 (directed against an N-terminus epitope) and JH455 (directed against a C-terminus epitope) showed complete colocalization of labelling (not illustrated). This is evidence that a full-length, functional S opsin is present. Interestingly, a very large proportion of aardvark cones throughout the retina showed co-expression of the L and S opsin. This is detailed below.

For aardvark 1, cone densities were assessed in many sample fields in flat-mounted pieces from various positions across central and peripheral retina. The positions of the counting regions are indicated in the retina scheme S1 Fig, and the cone density data are given in S2 Table. The most detailed counting was done along a vertical strip located about 5mm temporal of the optic disc and running from the dorsal edge of the retina through the streak to the ventral midperipheral retina (region 1 in S1 Fig). The results are presented in Fig 8 and show a clear centro-peripheral density gradient with a peak density of 1276 cones/mm² in the region of the horizontal streak, and lowest densities of below 400 cones/mm² in dorsal far periphery. In nasal retina, however, cone densities close to the streak were 530-630/mm² (regions 2, 3 in S1 Fig; see S2 Table), not different from or lower than cone densities in dorsal midperiphery (region 4) or dorsal periphery (regions 5-7). Ventral far peripheral retina had particularly high cone densities of around 900/mm² (region 8). In a large region of temporal retina (region 9), cone densities ranged from 1072/mm² in the streak to about 600-900/mm² at positions a few millimetres away in dorsal and ventral directions. In the counts of this temporal region we did not see a local cone density peak that would be indicative of a central area.

Overall, the temporal streak region showed a clear density peak, but elsewhere in the retina, there was no strong dependence of cone density from retinal location (S2 Table). With rod densities of up to 214,000/mm² in central and down to 124,000/mm² in peripheral retina, the above cone densities correspond to cone percentages of 0.25 – 0.85% among the photoreceptors, with a rough average of about 0.5% cones over most of the retina. For aardvark 2, total cone densities could only be determined in the two small peripheral pieces of retina described above. They ranged from 1,100 to 1,600 cones/mm². These densities are somewhat higher than those of aardvark 1. With the higher rod densities of 163,000 – 245,000/mm², this again amounts to cone percentages of 0.5 – 1.0% of the photoreceptors.

In sample fields across the retina of aardvark 1, between 35% and 98% were dual pigment cones (S2 Table). A substantial proportion of the cones were pure S cones without L opsin expression (some marked in Fig 7A–C). In dorsal retina, between 2% and 37% of the cones were pure S cones, in ventral retina, this percentage was between 23% and 65% (Fig 8, S2 Table). Hence, it appears that the proportion of pure S cones is markedly higher in ventral than dorsal retina, but the large variation of percentages between sampling fields also indicates local variability. Only a small proportion of the cones were pure L cones without S opsin expression. In many counting fields containing between 100 and 300 cones, pure L cones were absent; in other fields, pure L cones comprised 0.3% to 7% of the cones with no obvious regional trend.

In the retina of aardvark 2, the proportions were somewhat different. In several counting fields from a sample region in far peripheral retina, 67-80% of the cones were co-expressing L and S opsin, 15-21% were pure L cones, and 6-12% were pure S cones. In another region from an unknown but probably also relatively peripheral location, 86-95% of the cones were co-expressing L and S opsin, 2-6% were pure L cones, and 3-8% were pure S cones (S2 Fig).

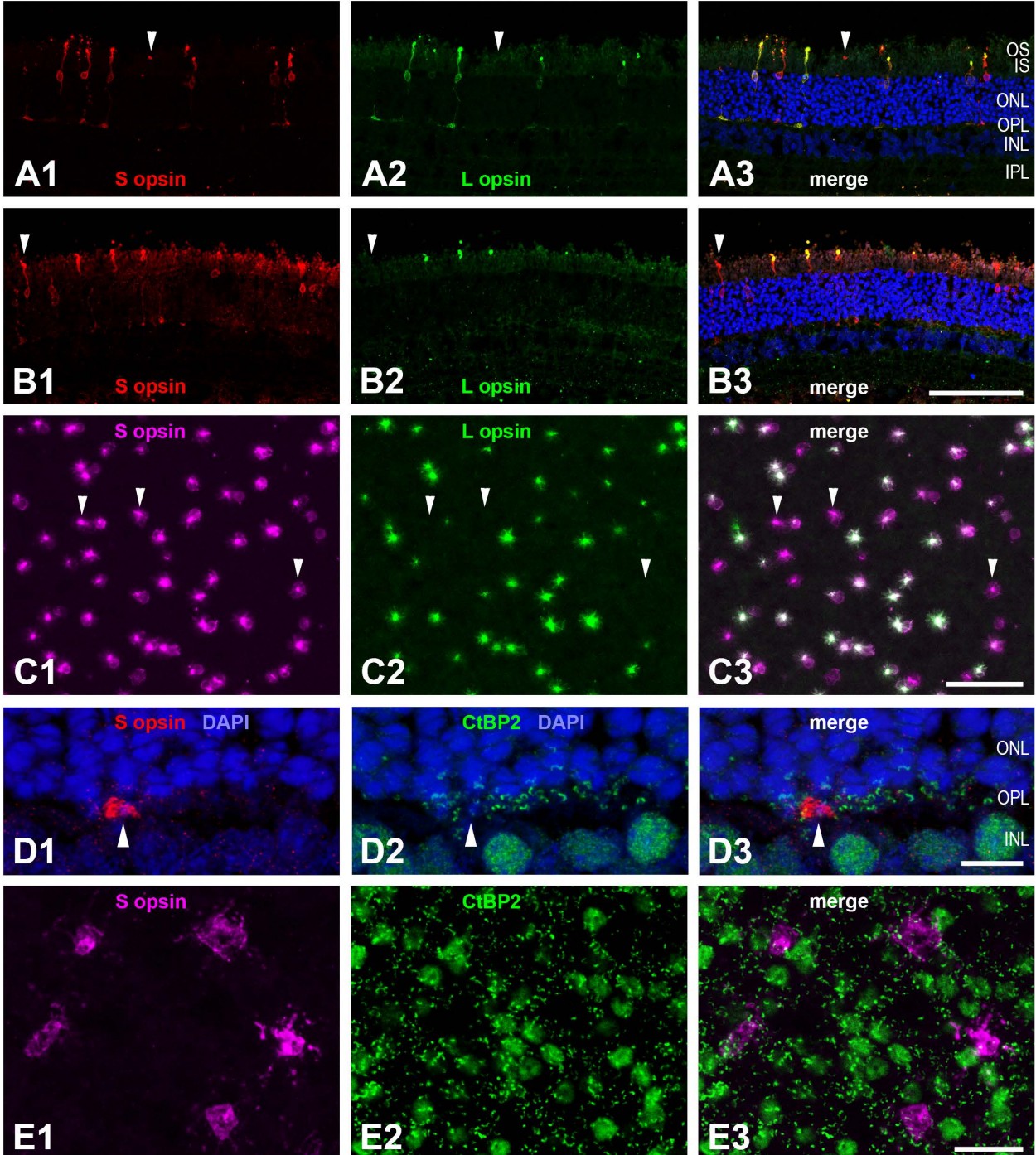

**Fig 7. Cone photoreceptors (aardvark 1).** (A, B) Vertical retinal sections double-immunolabelled for S cone opsin (A1, B1, red) and L cone opsin (A2, B2, green). In the merged images, DAPI counterstaining in blue shows the retinal nuclear layers (A3, B3). Most cones co-express S and L opsin, arrowheads point to pure S cones. In many cones of the region shown in (A), both the S opsin and the L opsin label extend throughout the cone from the OS to the cone pedicle in the OPL; in the region shown in (B), the L opsin label is restricted to the OS. (C) Double immunola-belled cones in a flat-mounted piece from ventral midperipheral retina. S opsin label (C1, magenta) and L opsin label (C2, green) are colocalized in most cones (C3). Three of the pure S cones are indicated by arrowheads, pure L cones are not present in this field. The aureole seen around most cones is an artefact; in this image the cones were photographed from the vitreous side of the retina, hence there is light scatter at many cellular structures. (D) Vertical section double-immunolabelled for S opsin (D1, in red, showing an S cone pedicle in the OPL) and the synaptic ribbon marker CtBP2 (D2, in green). Most of the small CtBP2 structures in the OPL are ribbons of rod spherules; the merge (D3) shows that cone pedicles do not have the ribbon/CtBP2 clusters seen in other mammals. As expected, many somata in the INL are also CtBP2-labelled. (E)

Flat-mounted retinal piece double-labelled for S opsin (E1, magenta) and CtBP2 (E2, green). The focus is on the cone pedicles in the OPL. The merge (E3) confirms the absence of cone-typical ribbon/CtBP2 clusters at the cone pedicles. (A-E) are maximum intensity projections of confocal image stacks. The stack in (C) starts at the level of the intensely labelled cone outer segments and ends at the cone soma level. In this region, faint S opsin label extended throughout the cone, whereas L opsin label was restricted to the outer segment in most of the cones. The stack in (E) is of 3 focal images spaced 0.5 μm apart. As this retinal piece was not perfectly flat, in some places the stack included the CtBP2-labelled INL somata (large green blobs, also seen in D) and missed some ribbons located in the outer OPL (seemingly ribbon-free patches). For layer abbreviations, see Fig 5. Scale bar in (B3) is 100 μm and applies to (A, B); scale bar in (C3) is 50 μm; scale bar in (D3) is 10 μm; scale bar in (E3) is 20 μm.

These percentages have to be considered with some reservation. First, the relative intensity of the L and S opsin label varied considerably between cones, in some cones one of the labels was barely above background. This was particularly obvious in aardvark 2 (S2 Fig) but can also be seen in aardvark 1 (Fig 7C). Hence, assessment of opsin co-expression has a subjective component. Second, in many retinal regions, the L opsin label was restricted to the cone outer segments (Fig 7B), whereas the S opsin label typically extended throughout the cone including the soma, axon, and cone pedicle (Fig 7A,B,D,E). The cone opsin labelling revealed some degree of post mortem outer segment damage, some outer segments were elongated, others were just small stumps (see Fig 7C). Hence, S opsin-labelled cones could be identified rather reliably by focusing through the outer retina to their soma level, whereas L opsin expression may have been missed in cases of outer segment loss. Nevertheless, there were many sampling fields where the cone outer segments were partially preserved, so the above proportions of pure S and L cones are at least semi-quantitatively reliable. In the temporal retina of aardvark 1 (region 9 in S1 Fig), for technical reasons the L opsin label was faint, hence the proportions of dual pigment cones and thus also pure S cones could not be reliably determined and are not given in S2 Table. As pure L cones could be identified due to their higher L opsin content, the total number of cones (i.e., S opsin-expressing cones and pure L cones) could be determined.

The synaptic ribbon marker CtBP2 mainly revealed the single synaptic ribbons of rod synaptic endings (rod spherules) in the OPL (Fig 7D and E). These synaptic ribbons often showed the horseshoe shape that is typical for mammalian rod ribbons (Fig 7D2). In contrast to the situation in other mammals, CtBP2 did not show the typical clustering of ribbons in the cone pedicles; there were only a few or no CtBP2 puncta at the pedicles of S cones (Fig 7D and E).

The general cone marker peanut agglutinin (PNA) labelled the S cone pedicles in the aardvark retina (S3 Fig). This was evident by the overlapping label of the pedicles by the S opsin antiserum and PNA. The lateral displacement of the pedicles against the cone outer segments (S3C Fig) is mostly due to slight tissue shearing during the mounting of the retinal pieces and could be followed by tracing the stained S cone axons through the ONL in the image stacks. The PNA label of cone pedicles varied in intensity and size; in a few S cone pedicles it was absent or too faint to be detected. Surprisingly, the L cone pedicles did not show PNA labelling. This was checked in several L cones found in appropriately stained retinal pieces. Two pure L cones are present in the field shown in S3 Fig. It is unknown whether PNA does not label the L cone pedicles at all, or whether the label is below the detection threshold of our staining.

### Thyroid hormone levels

Given the high incidence of cone opsin co-expression in the aardvark and the important role of thyroid hormone (TH) in regulating cone spectral identity, we looked at the serum TH levels available for aardvark 1 (five measurements within 2.5 years) and his adult son aardvark 3 (one measurement) in comparison to published serum TH levels in some representative mammals (Table 2). To our knowledge, these are the first published serum TH levels for aardvarks. The serum values of total thyroxine (tT4), total triiodothyronine (tT3), free T4 (fT4)

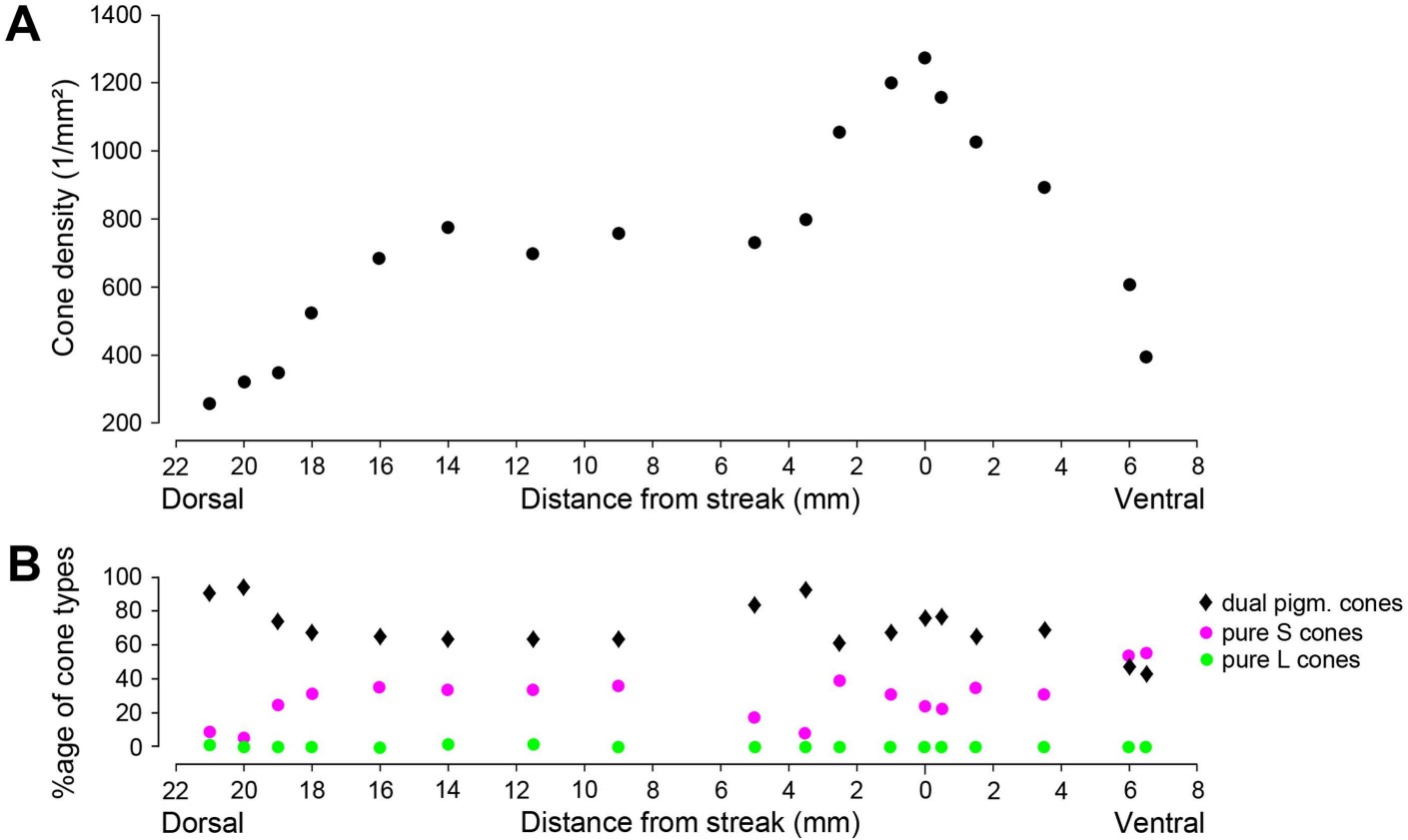

**Fig 8. Cone densities and opsin expression patterns along a dorso-ventral strip in temporal retina (aardvark 1).** (A) Cone density along the dorso-ventral strip (region 1 in the retina scheme S1 Fig) decreases with distance from the streak, the horizontal band of reduced pigmentation in the fundus (see Fig 3). (B) Percentages of dual pigment cones (black diamonds), pure S cones (magenta dots) and pure L cones (green dots) along the dorso-ventral strip. The strip runs from the dorsal edge of the retina (21 mm dorsal) to ventral midperipheral retina (7 mm ventral). Counting field sizes were 500 μm x 500 μm.

and the biologically active form free T3 (fT3) are very similar for aardvark 1 and his son, and are also similar to or higher than the respective values in other mammals. This suggests that aardvarks are not hypothyroid and that the S opsin dominance is not related to a lack of TH (see Discussion).

## Discussion

The aardvark's designation as a 'living fossil' [4] suggests that its eye and retina also may show prototypic, primordial mammalian features. On the other hand, the aardvark ancestor could already have been specialized on feeding on ants and termites, which speaks against a 'basic general' mammal. Hence, we were interested to study the aardvark's eye and retina when the opportunity arose. Early mammals are assumed to have gone through a 'nocturnal bottleneck' with associated adaptations to low-light vision (see, e.g., [32–36]). Most extant nocturnal mammals possess eye and retina features reflecting such an evolutionary adaptation: The eye optics with a proportionately large lens and cornea serves increased light capture. The retina has a dominance of the more light-sensitive rod photoreceptors and only a small minority of cone photoreceptors (review: [7]). The features of the aardvark eye and retina fit this 'nocturnal' category.

Until 2022, the only detailed study available on the aardvark eye and retina was by Victor Franz, dating back to 1909 [5]. It claimed a pure rod retina without cones, but concedes

**Table 2. Thyroid hormone serum levels in different mammals. (Different units in the publications have been converted to unify the table).**

| Species/ Animal | tT4 (μg/dl) | tT3 (ng/dl) | fT4 (ng/dl) | fT3 (pg/ml) | TSH (ng/ml) |
|---|---|---|---|---|---|
| Aardvark 1[1] | 12–>15 | 226.3–390.4 | 1.54–1.93 | 2.28–4.23 | <0.03 |
| Aardvark 3[1] | >15 | 304.2 | 2.25 | 3.32 | nd |
| Asian elephant[2] | 11.12 | 126.72 | 0.87 | 1.39 | 0.97 |
| African elephant[2] | 10.76 | 123.27 | 0.93 | 1.41 | 0.56 |
| Manatee[3] | 4.5–8.3 | 140–160 | 1.3–1.6 | nd | nd |
| Cow[4] | 3.9–8.8 | 70–250 | 1.3–2.4 | 2.3–7.8 | nd |
| Dog[5,6] | 1.5–1.6 (1.53–2.25) | 76–84 (nd) | ~1.75 (0.98–1.57) | ~1.15 (nd) | 0.10-0.18 (0.07–0.26) |
| Human[7] | 5–12 | 80–220 | 0.7–1.9l | 2.3–4.1 | 0.006–0.06 |

tT4 = total thyroxine T4, tT3 = total triiodothyronine T3, fT4 = free T4, fT3 = free T3; TSH = Thyroid-stimulating hormone, thyrotropin; nd = not determined.

[1]Frankfurt Zoo animals (see Methods);

[2]Ref. [27];

[3]Ref. [28];

[4]Ref. [29];

[5]Ref. [30];

[6]values in brackets, Ref. [31];

[7]UCLA Health Website (https://www.uclahealth.org/medical-services/surgery/endocrine-surgery/conditions-treated/thyroid/normal-thyroid-hormone-levels), fT3 level from Cleveland Clinic (https://my.clevelandclinic.org/health/diagnostics/22425-triiodothyronine-t3)

that the tissue conservation may not have been sufficient to prove the absence of cones. The absence of cones has since been repeated in various printed summaries and websites on aardvark biology and vision ([37] page 326, [38] page 579, [3,39,40]; website examples: [41,42]). Franz [5] also claimed the absence of a tapetum lucidum, but conceded in a later handbook chapter that a tapetum lucidum fibrosum may be present ([43] page 1214). Recently, Paszta and colleagues published a study on general features of the aardvark eye and extraocular structures that included a brief description of the retina [11]. This paper similarly claimed an absence of cones, but reported a tapetum lucidum. Walls [32] also listed the aardvark as having 'eye shine' and a trace of a tapetum fibrosum (his Table VII, p. 241). The present study has assessed these claims.

The external dimensions of the eye of aardvark 1 are similar to those reported previously [5,11]. The corneal size and lens are large compared to total eye size, the ratios of corneal diameter to eye equatorial diameter (0.75), of corneal diameter to eye axial length (0.81), of lens diameter to eye equatorial diameter (0.58), and of lens thickness to eye axial length (0.45) are within the range seen across nocturnal mammals [34,44]. However, these parameters overlap between nocturnal, cathemeral/crepuscular, and diurnal mammals, so their diagnostic value is limited (for discussion see, e.g., [34], but also [45]).

## Tapetum lucidum

Some nocturnal, crepuscular, and arrhythmic mammals have a reflective choroidal tapetum lucidum behind the retina to increase the amount of light absorbed by the photoreceptors. To observers the tapetal reflection appears as 'eye shine.' Morphologically, the two most common tapetum types are a 'tapetum cellulosum' where the reflecting structures termed rodlets are located intracellularly (found in Carnivora), and a 'tapetum fibrosum' where the reflecting structures termed fibrils are located extracellularly (found in Artiodactyla, Cetacea and Perissodactyla) (reviews: [46–48]). Our observations (Fig 2) indicate that the aardvark may have a tapetum lucidum. Hence, it is surprising that Franz [5], Matas et al. [49] and Freeman et al. [2] describe

the absence of a tapetum in the aardvark eye. In contrast, Walls [32] and Paszta et al. [11] describe the presence of a tapetum, which they consider to be a choroidal tapetum fibrosum.

Our more detailed histological observations confirm a choroidal tapetum. The ultrastructural appearance of the tapetal fibrils with their cross-striation (Fig 4C) indicates that the fibrils consist of collagen, which is typical for a tapetum fibrosum [47,50]. On the other hand, in cross-section, the aardvark tapetal fibrils show a substructure with concentric zones of different electron-density (Fig 4D). This is reminiscent of the substructure of the zinc-containing rodlets of the tapetum cellulosum in ferret and dog [51,52]. As our tissue fixation was not optimized for electron microscopy, the substructure of the aardvark tapetal fibrils may be an artefact. Unfortunately, our material did not allow us to determine whether the domains with their changing fibril orientations are contained intracellularly (suggesting a tapetum cellulosum) or whether they are extracellular (suggesting a tapetum fibrosum). Weighing all evidence, we think that the aardvark tapetum lucidum is of the fibrosum type. We consider it unlikely that the aardvark has a hybrid tapetum with a mix of fibrosum and cellulosum features, because to date that has not been reported in any mammal. Ultrastructural studies on suitably preserved tissue will be needed to provide a definitive answer. In mammals with a tapetum lucidum (whether of the cellulosum or fibrosum type), the RPE cells in front of the tapetum are unpigmented or only minimally pigmented, such that the tapetum can in fact function as a reflector (reviews: [32] page 232; [47]). This is also the case in the aardvark.

Across mammals, the optically relevant properties of tapetum fibrosum fibrils and tapetum cellulosum rodlets are similar. Both have diameters of about 100-200 nm and are nearly hexagonally arranged with a spacing of about one fibril/rodlet diameter, suitable for constructive interference (e.g., sheep: [50]; bovine: [53]; cat: [54,55]; overviews: [47,48,56]). The aardvark fibrils have comparable dimensions, but their spacing is more disordered. In many domains they are loosely spaced and do not show a tight hexagonal arrangement. Also, the aardvark tapetum layer is relatively thin and the tapetum lamination is not as conspicuous as in other species. Overall, the aardvark appears to have a rudimentary version of a tapetum lucidum. Nevertheless, it obviously shows the typical 'eye shine.'

## Avascular retina

The aardvark retina is avascular, as observed by Franz [5] and Matas et al. [49], and confirmed here. This also explains why we did not see any GFAP-labelled astrocytes. Retinal astrocytes are neuroglia cells restricted to the optic nerve fibre layer and associated with blood vessels as well as with retinal ganglion cell axons. In species with limited retinal vascularization, they only occur in the vascularized regions, and in avascular retinae they are completely absent (review: [21]). Avascular retinae have been found in several mammals from different orders [32,57–60]. There is no convincing common explanation for why the retinae of some species are avascular whereas those of most other species are vascularized to various extents. Damsgaard and Country [60] conclude from their large data survey that the ancestral mammal had an avascular retina, and that retinal vascularization was dynamically gained and lost throughout subsequent mammalian evolution depending on species-specific visual needs for retinal processing capacity, neuron numbers and hence retinal thickness. Mammalian avascular retinae are generally thinner than vascularized ones, commonly below 150 μm (reviews: [59,60]). The reason is that they have to be supplied with oxygen by diffusion from the choroidal capillary network, and the maximum oxygen diffusion distance has been modeled to be about 143 μm [61], although this value has been questioned by some authors [62]. At 120-180 μm, the aardvark retina is slightly thicker than other avascular retinae, but it is still thinner than many vascular nocturnal retinae, particularly having a thinner ONL and lower photoreceptor density (see below).

Chase [59] also stated that mammals with avascular retinae lack a tapetum lucidum and assumed that this is because a tapetum would add to the tissue thickness that has to be reached by choroidal oxygen. The aardvark tapetum lucidum is not in line with that assumption. The elephants, the horse and the zebra are further examples of species with a tapetum lucidum and an avascular retina [57,58]. In fact, the choriocapillaris, the actual release site of oxygen and nutrients to the retina, is located in front of the choroidal tapetum, directly facing the RPE and retina. Fig 4E–H shows this for the aardvark and elephant. The aardvark choriocapillaris forms a very dense capillary mesh (Fig 4I) that obviously suffices to adequately supply the retina. We conclude that Chase's assumption of an incompatibility of a (choroidal) tapetum lucidum with an avascular retina is not tenable.

## Rods

Typically for a nocturnal mammal, the vast majority of the aardvark photoreceptors are rods; the proportion of only around 0.5% cones is low even among nocturnal species (review: [7]). However, the estimated rod densities of 124,000 – 214,000/mm² for aardvark 1 and of 163,000 – 245,000/mm² for aardvark 2 are rather low in comparison to those of many other nocturnal mammals, which may range from 200,000 rods/mm² to more than 700,000 rods/mm² [7]. These low rod densities and the correspondingly thinner ONL most likely are correlated with the avascularity of the aardvark retina. The nocturnal Microchiroptera also have avascular retinae (review: [59]), and rod densities in the greater horseshoe bat average about 370,000 rods/mm², complemented by about 2.5% cones [63]. The avascular retina of the nocturnal to crepuscular rabbit has between 300,000 rods/mm² in central and 130,00 rods/mm² in peripheral retina, complemented by about 4 – 6% cones [64]. Hence, even among nocturnal mammals with avascular retinae, the aardvark rod density is in the lower range, suggesting a low adaptive pressure for good nocturnal vision.

Also typical for nocturnal mammals is an inverted architecture of the rod nuclei [24]. Franz [5] noted the central clustering of heterochromatin in the aardvark rods and described it as 'nuclei with two opposing chromatin bodies that looked like being in mitosis.' We here give a detailed description of these rod nuclei (Fig 6). As analyzed by Solovei and colleagues [24,26], the central position of the inactive heterochromatin and peripheral position of the active euchromatin are assumed to be strongly disadvantageous for nuclear functions, but they have an optical advantage. The densely packed heterochromatin core is highly refractive and shows the physical properties of a light-focusing lens. Hence, the rod nuclei in the ONL form columns of microlenses that act as 'light guides.' This strongly reduces light scattering in the ONL, which is particularly important for nocturnal mammals with their thicker ONL and the need to capture a large proportion of the few photons available at night. Obviously, inverted rod nuclei are an evolutionary adaptation to improve nocturnal vision. The rods of diurnal mammals have a conventional nuclear architecture, because they do not have to maximize photon capture.

The aardvark has semi-inverted rod nuclei. Most likely, these are less effective focusing lenses than fully inverted rod nuclei. Both the relatively low rod density compared to other nocturnal mammals and the semi-inverted rod nuclei support the assumption that for the aardvark with its reliance on smell and hearing, there was no strong evolutionary pressure for an optimally adapted nocturnal retina, and that its night vision sensitivity is lower than that of many other nocturnal mammals.

In conventional nuclei, heterochromatin is tethered to the nuclear envelope by either lamin A/C or the lamin B receptor (LBR); the inverted rod nuclei of nocturnal mammals are characterized by the absence of both tethers [65]. The semi-inverted architecture of the aardvark rod nuclei might be explained by expression of one of these proteins. We tested this

with antibodies to LBR and to lamin A/C. Whereas no aardvark retinal neurons showed LBR immunoreactivity, all of them showed lamin A/C immunoreactivity. The lamin A/C antibody clearly marked the rod's nuclear periphery, although admittedly weaker compared to the nuclei of the INL and GCL. This is similar to the weak LBR expression in the semi-inverted rod nuclei of, e.g., goat and cow [65].

## Cones

From today's perspective, the claim that the aardvark retina completely lacks cones [5,11] was based on histological approaches that are not suitable to detect sparse populations of cones. The most reliable approach to identify cones is by immunolabelling them with antibodies to cone opsins, which we have done in the present study. The immunolabelling showed the presence of L and S opsin in a low-density population of aardvark cones.

The two common mammalian cone opsins are the L opsin (also termed LWS opsin, peak sensitivity $\lambda_{max}$ in the green to yellow part of the spectrum) and the S opsin type 1 (SWS1 opsin, $\lambda_{max}$ in the blue, violet or UV part of the spectrum [66]. The opsin antisera used here recognize all spectral variants of the respective opsins. Partial gene sequencing has identified L and S opsin genes in the aardvark, and judging from the tuning-relevant amino acids, the aardvark S opsin has been suggested to be UV-sensitive, and the L opsin to have its $\lambda_{max}$ at 527-533 nm (Supplementary Table S5 in [9]). UV tuning is the ancestral tuning of the verte-brate SWS1 opsin and has been maintained in a number of nocturnal mammals (reviews: [66–69]). Our opsin immunolabelling gives evidence that the L and S opsins are indeed expressed in aardvark cones, and provides information about the population density and topographical distribution of the cones.

A less normal feature of the aardvark cones is, that many of them co-express the two opsins across the retina and are 'dual pigment' cones. An artefactual double-labelling due to cross-reactivity between the two opsin antisera can be excluded, because the tissue also shows pure L and S cones. Many nocturnal mammals have the two opsins in separate cone populations and thus the basis for dichromatic color vision. This is the canonical mammalian condition of being spectrally selective (reviews: [6,7,70]). Dual pigment cones were reported in the ventral retinae of the house mouse, guinea pig and rabbit [71,72]. Since then, opsin co-expression in all cones across the retina has been found in the pouched mouse *Saccostomus campestris* and the Siberian dwarf hamster *Phodopus sungorus* [73]. The molecular mechanisms responsible for the co-expression of both opsins in some cones and its suppression in other cones is only partly understood. During rat and gerbil retinal development, all cones first express S opsin, and prospective L cones then successively switch to L opsin expression with an intermediate phase of opsin co-expression; this suggests that S opsin expression may be the default path-way when an L opsin-activating mechanism is absent or suppressed [74]. One such factor is thyroid hormone (see below).

An evolutionary comparison to the photoreceptor properties in other Afrotheria is limited by the paucity of published data. In the retina of the nocturnal lesser hedgehog tenrec *Echinops telfairi* (Tenrecidae, see Fig 1), the average rod density is 255,000/mm², and cone densities increase from 3000-3700/mm² in dorsal to around 7,000/mm² in ventral retina, comprising 1.5-2.3% of the photoreceptors [75]. The S cone proportion among the cones increases from 10-30% in dorsal to 40-70% in ventral retina; the S cones outnumber the L cones over a signif-icant part of ventral retina, but there are no dual pigment cones [75]. In the other main branch of the Afrotheria, some cone data are available for the African elephant *Loxodonta africana* and the West Indian manatee *Trichechus manatus* (Proboscidea and Sirenia, respectively; see Fig 1). The African elephant has both L and S cones [16], and there is no co-expression of the two cone opsins [76]. The elephant L opsin has its $\lambda_{max}$ at 552 nm, and the S opsin at

419 nm (violet) [77]. The West Indian manatee has L opsin and S opsin, with $\lambda_{max}$ at 555 nm and 410 nm (violet), respectively [78]. There are no immunohistochemical data on the opsin expression pattern (pure L and S cones, or co-expression in some cones) and the cone density distribution across the retina.

If a conclusion can be drawn from these four studied animals (aardvark, tenrec, elephant, manatee), it is that the cone properties, proportions and distributions probably are as diverse across Afrotheria as they are across mammals in general. The L and S opsin may be expressed in separate cones (tenrec, elephant) or co-expressed in many cones (aardvark), the S cone proportion may markedly differ between dorsal and ventral retina (tenrec) or stay rather constant across the retina (elephant; L. Peichl, unpublished observation), and the S opsin may be tuned to UV (aardvark) or to violet/blue (elephant, manatee).

In the retina of aardvark 1, the majority of single pigment cones are pure S cones (regionally varying from 2% to 65% of the cones); pure L cones are a sparse population of about 0.3 – 7%. In the peripheral retina of aardvark 2, there are somewhat more pure L cones (2 – 21%) than pure S cones (3 – 12%). If the aardvark has colour-processing (cone-opponent) retinal ganglion cells, most of their input will be from dual pigment cones. It is possible that the aardvark uses the spectrally different sensitivities of pure L and S cones vs. dual pigment cones for colour vision. The rods may also contribute to colour vision in mesopic light conditions [66,79]. However, given the very low cone density and the dominance of dual pigment cones, it is likely that the aardvark at best has feeble colour vision [70]. The low cone density also makes good photopic visual acuity (cone-based spatial resolution) unlikely.

Our opsin immunolabelling showed qualitatively different mixtures of the two opsins in different cones, but did not allow to quantitatively access the absolute amounts of the two opsins per cone. Hence, we cannot comment on the presumed dominant spectral sensitivity of these dual-pigment cones. However, the much higher fraction of S opsin-containing cones in comparison to most other mammals suggests that the aardvark retina has a relatively high cone-based sensitivity in the shortwave (blue to UV) range. This is the case, e.g., in Microchiroptera with a similar S opsin dominance [80]. It may be an evolutionary adaptation to the spectral composition of twilight, which contains higher proportions of short wavelengths than full daylight (see, e.g., [81,82]). Twilight is the most likely situation that nocturnal animals may encounter during their active phases, and cones contribute to vision at this mesopic light level. On the other hand, many nocturnal mammals facing the same twilight conditions, e.g., rats [83], flying foxes [84], colugos [18], nocturnal lemurs [85], and some Canidae [86], have low proportions of S cones (reviews: [6,7,68]). Moreover, a substantial number of nocturnal mammals are completely lacking S cones (review: [87]). That makes the hypothesis of a special shortwave adaptation to twilight less plausible.

As assessed in the retina of aardvark 1, cone density changes across the retina are relatively small, ranging from a maximum of up to nearly 1,300 cones/mm² in the streak region of temporal retina to minima of about 300 cones/mm² at some peripheral locations, but up to 950 cones/mm² in other peripheral regions, representing 0.25–0.85% of the photoreceptors. The temporal part of the streak (horizontal band of reduced RPE pigmentation seen in the eye's fundus) has a higher cone density than the surrounding midperiphery, but the nasal part of the streak does not. The cone topography does not suggest specialized regions of particularly high cone-based visual performance like a prominent area centralis or visual streak, but a full cone density map of the retina would be needed to draw firm conclusions. Such specialisations are present in many other mammals (reviews: [88–90]), but often most prominent in the retinal ganglion cells.

The cone density in the small parts of the retina of aardvark 2 that could be studied is slightly higher, but due to the higher rod densities there, the cones also represent only

0.5–1.0% of the photoreceptors. Together with the different proportions of pure L cones, pure S cones and dual pigment cones in the two individuals, it appears that there is some interindividual variability in the detailed characteristics of the cone population, even though the basic properties of a low cone-to-rod ratio and a dominance of dual pigment cones are preserved. Alternatively, these differences between the old male aardvark 1 and the younger female aardvark 2 could be sex-related or age-related. A larger sample of aardvark retinae would be needed to study these aspects.

A further unusual feature of the aardvark cones is that the cone pedicles do not show the typical cluster of CtBP2-positive ribbons. In other mammals, the cone pedicles have a large number of presynaptic sites with ribbons (reviews: [91]–[93]). The tissue quality did not allow an ultrastructural analysis, so we could not determine whether the aardvark cones have only few ribbon synapses per pedicle, or whether there are many that do not label for CtBP2. A low ribbon density may impair the signal transmission performance of the aardvark cones. In addition to the ribbons, many INL somata were CtBP2-labelled. This has also been seen in other mammals, e.g., in mouse [91] and bat [94].

## Thyroid hormone (TH)

In mouse early postnatal development, TH, through its receptor TRβ2, is a crucial regulator of cone spectral identity by repressing S opsin and activating L opsin [95]–[97]. Even in adult mouse and rat, pharmacological suppression of serum TH reversibly activates S opsin and represses L opsin in all cones [98]. It may be that dual pigment cones are the consequence of a (genetic) defect of the switch-off mechanism for S opsin expression during developmental L opsin activation [99]. Hence, we were interested to know whether the aardvark has unusually low TH levels that might correlate with the high proportion of dual pigment cones. The data presented in Table 2 show that this is not the case, the aardvark has serum TH levels that are similar to or higher than those of other mammals, particularly those of the elephants and the manatee that belong to the same clade Afrotheria as the aardvark. The other listed species cover a broad range of mammals, and all of them have a normal cone complement with a majority of pure L cones and an approximately 10% minority of pure S cones (with the exception of the manatee, for which no cone population data exist). This suggests that aardvarks are not hypothyroid and that the S opsin dominance is not related to a lack of TH. Also, elevated serum levels of thyroid-stimulating hormone (TSH) would be a first sign of hypothyroidism, but they appear low in aardvark 1 (Table 2). However, there could be other deficits in the chain of thyroid hormone action that we were unable to study here. For example, one crucial component is the nuclear T3 receptor TRβ2; in TRβ2 knockout mice, all cones express S opsin and none express L opsin [95]. Another component is the monocarboxylate transporter 8 (MCT8), a plasma membrane transporter allowing TH access to the cones; in MCT8 knockout mice, cone opsin expression resembles that in hypothyroid or TRβ2 knockout mice [100].

## Conclusions

The aardvark eye and retina have the typical features seen in nocturnal mammals: light-sensitive optics with a large lens and a large cornea, a reflective tapetum lucidum, and a rod dominance with a very low cone density. Three unusual features are, (i) that the retina is avascular with the corollary of being thinner and hence having a lower rod density than many other nocturnal mammals, (ii) that the rod nuclei are only semi-inverted and not fully inverted as in other nocturnal mammals, (iii) that there is opsin co-expression in a large proportion of the cones. Hence, nocturnal visual sensitivity probably is lower than in many other nocturnal mammals, and cone-based visual acuity and colour vision certainly are poor.

Regarding the designation of the aardvark as a living fossil [4] and the question whether this entails a primordial, prototypical mammalian retina, one can state that the massive presence of dual pigment cones argues against a primordial character of the aardvark retina. The most common pattern in non-mammalian vertebrate retinae, as in mammalian retinae including those of Afrotheria, are single opsin cones. The most parsimonious assumption is that the last synapsid ancestor of the mammals, and hence also the first mammal, had single opsin cones. Therefore, dual pigment cones are a derived feature, and the aardvark retina is not a primordial mammalian retina as far as the cones are concerned.

## Supporting information

**S1 Fig. Sampling regions for photoreceptor densities** . Schematic drawing of the right retina of aardvark 1, not drawn to scale. The thick black vertical line shows the dorsal strip that was used for transverse retinal sections to assess photoreceptor (rod) densities in the outer nuclear layer (see S1 Table). The rectangles 1-9 are the regions where cone densities were accessed in several counting fields. Cone densities along the dorso-ventral strip 1 are shown in Fig 8, cone densities in regions 2-9 are given in S2 Table. The central grey oval represents the optic disc, the broken horizonal line represents the bright streak of reduced fundus pigmentation (see Fig 3C and D). T, temporal; N, nasal; D, dorsal; V, ventral.
(TIF)

**S2 Fig. Cone photoreceptors** . Sequentially double immunolabelled cones in two neighbouring flat-mounted pieces from an unknown location in midperipheral to peripheral retina (aardvark 2). The piece of field 1 was first incubated with the rabbit L opsin antiserum JH492, visualized with an Alexa488-coupled secondary antiserum (green). Then it was incubated with the rabbit S opsin antiserum JH455, visualized with a Cy3-coupled secondary antiserum (magenta). The merge shows that all cones are labelled by Cy3, because this secondary antiserum bound to JH455 as well as JH492. The pure S cones are exclusively labelled by Cy3 (arrowheads), all Alexa488-labelled cones contain the L opsin. The neighbouring piece of field 2 was first incubated with the rabbit S opsin antiserum JH455, visualized with the Cy3-coupled secondary antiserum (magenta). Then it was incubated with the rabbit L opsin antiserum JH492, visualized with the Alexa488-coupled secondary antiserum (green). The merge shows that all cones are labelled by Alexa488, because this secondary antiserum bound to JH492 as well as JH455. The pure L cones are exclusively labelled by Alexa488 (arrowheads), all Cy3-labelled cones contain the S opsin. The relative amount of L and S opsin (i.e., labelling intensity) differs between cones. For details of the procedure see Methods. The images are maximum intensity projections of confocal image stacks. Scale bar is 50 μm and applies to all images.
(TIF)

**S3 Fig. Cone photoreceptors and their synaptic pedicles in the outer plexiform layer** . Flat-mounted piece of retina (aardvark 1), triple immunolabelled for S opsin, L opsin and PNA. (**A**) Focus on the outer segments of the opsin-immunolabelled cones; most cones express both opsins (A1, S opsin; A2, L opsin; A3, merge). The field contains two pure L cones. (**B**) Focus on the cone pedicles in the outer plexiform layer; the S opsin-labelled pedicles (B1) are also labelled by PNA (B2), as shown in the merge (B3). (**C**) Schematic illustration of the cones identified in A (S cones, magenta circles; L cones, green circles; dual pigment cones, bipartite circles) and the PNA-labelled pedicles identified in B (grey squares). The pairing between outer segments and pedicles (connecting lines) was checked by following the S opsin-labelled cone axons through the image stacks. The pedicles of the two pure L cones show no PNA label. For two pedicles at the edges of the image, the corresponding outer segments lie outside

the frame. Due to faint labelling of some cones, not all cones shown in (C) can be seen in (A) and (B). Scale bar in (C) is 50 μm and applies to all images.
(TIF)

**S1 Table. Aardvark photoreceptor (rod) densities.**
(DOCX)

**S2 Table. Cone densities in various retinal regions of aardvark 1.**
(DOCX)

## Acknowledgments

We thank Robert Molday and Jeremy Nathans for kindly providing antibodies. The technical assistance of Alena Konoplew, Elke Laedtke and Carola Tröger is gratefully acknowledged. We also thank Radosław Ratajszczak, Wojciech Paszta and Krzysztof Zagórski (Wrocław Zoo) for their help in collecting research material and for providing information on the animal from which the eyes were taken.

## Author contributions

**Conceptualization:** Leo Peichl.

**Investigation:** Leo Peichl, Sonja Meimann, Irina Solovei, Irene L. Gügel, Christina Geiger, Nicole Schauerte, Silke Haverkamp.

**Project administration:** Leo Peichl.

**Resources:** Christina Geiger, Nicole Schauerte, Karolina Goździewska-Harłajczuk, Joanna E. Klećkowska-Nawrot, Gudrun Wibbelt.

**Validation:** Leo Peichl.

**Visualization:** Leo Peichl.

**Writing – original draft:** Leo Peichl.

**Writing – review & editing:** Sonja Meimann, Irina Solovei, Irene L. Gügel, Christina Geiger, Nicole Schauerte, Karolina Goździewska-Harłajczuk, Joanna E. Klećkowska-Nawrot, Gudrun Wibbelt, Silke Haverkamp.

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
