## [Decision Letter · Decision Letter 0]

4 Dec 2024

PONE-D-24-50728Eye features and retinal photoreceptors of the nocturnal aardvark (Orycteropus afer, Tubulidentata)PLOS ONE

Dear Dr. Peichl,

Thank you for submitting your manuscript to PLOS ONE. After careful consideration, we feel that it has merit but does not fully meet PLOS ONE’s publication criteria as it currently stands. Therefore, we invite you to submit a revised version of the manuscript that addresses the points raised during the review process.

**The reviewers both think that the manuscript is well-written and of high standard. Both only had on minor revisions which should be easily to address. **

We look forward to receiving your revised manuscript.

Kind regards,

Gerrit Hilgen

Academic Editor

PLOS ONE

**Journal Requirements:**

Reviewers' comments:

Reviewer's Responses to Questions

**Comments to the Author**

1. Is the manuscript technically sound, and do the data support the conclusions?

Reviewer #1: Yes

Reviewer #2: Yes

2. Has the statistical analysis been performed appropriately and rigorously? 

Reviewer #1: N/A

Reviewer #2: Yes

3. Have the authors made all data underlying the findings in their manuscript fully available?

Reviewer #1: No

Reviewer #2: Yes

4. Is the manuscript presented in an intelligible fashion and written in standard English?

Reviewer #1: Yes

Reviewer #2: Yes

5. Review Comments to the Author

**Reviewer #1: ** This manuscript by Peichl L, et al. investigates the anatomical features of the nocturnal aardvark eye and its photoreceptor composition. The authors took advantage of a rare opportunity to obtain aardvark eye samples from a local zoo. Using standard modern techniques, they re-evaluated the findings made over a century ago, corrected previous misconceptions, and provided new descriptions. These include details on the structure of the tapetum lucidum, RPE pigmentation, rod density, rod nuclear structures, cone density, and cone pigment expression. This careful examination of these features provides valuable information about the species-specific adaptations of the eye in this ‘living fossil’ species, contributing to advancing our understanding of the evolutionary origins of the mammalian eye. The manuscript is well-written at a high standard, and the data are presented clearly, with a few exceptions noted in the minor comments below.

Minor comments

1. (Lines 443-445) ‘The observed range was about 124,000-214,000 photoreceptors (rods)/mm2, with densities decreasing from central to peripheral retina.’

Please provide the actual data showing the retinal locations and corresponding rod densities. This information is helpful in assessing how sharply or gradually rod density changes with retinal eccentricity.

2. (Lines 486-501)

The proportions of dual and single pigment cones are provided as ranges (e.g., between 35% and 96%). In some cases, the range is rather large. It would be helpful to include the actual data, regardless of the sample size, to know whether these variations are due to natural differences or a few outliers.

3. (Figure 6D-E) The resolution of these images (at least in the PDF version available for review) is insufficient to evaluate the descriptions provided in the text. For example, it is unclear how the authors concluded that there is only one synaptic ribbon in a rod spherule. Similarly, it is difficult to verify whether there are few or no ribbon at the S cone pedicles.

Additionally, the images (Fig. 6E2,E3) show large, intensely stained CtBP2 blobs in addition to punctate stainings. Are these blobs associated with rod spherules? Furthermore, there are apparent holes in the OPL where no CtBP2 labeling is visible (Fig. 6E2). Are these areas devoid of photoreceptor terminals?

**Reviewer #2:**  Given the unique chance of investigating the eye of the aardvark Orycteropus afer, Peichl et al. present their observations on eye morphology and retinal features. The study is valuable since the authors have for the first time thoroughly documented features of the aardvark eye with modern techniques, laying the basis for broader comparative anatomical comparison. Overall, the paper is well written, presents unique research, and reports results compellingly. The main recommendation is making the link to ancestral relationships and morphological evolution more cohesive. Changes to some figures are recommended. Other minor adjustments are also suggested.

The authors set an expectation of discussing their findings from an evolutionary perspective by describing aardvarks as “living fossils” (63-65) and evaluating the reported divergence from the mammalian scheme of having both rods and cones (72-73). Although compelling, the evolutionary implications are somewhat fragmented, and would be strengthened by including:

- A phylogenetic tree or another schematic (either in the introduction or the discussion) to support comparisons across species and demonstrate the aardvark’s relation to other species.

- Brief comparison of aardvark retina to model species in the discussion.

- A dedicated discussion section summarising the evolutionary implications of the findings, elaborating on the notion that the aardvark does not seem to have a primordial retina. This would reinforce the final paragraph in the conclusions (855-862).

The methodology section is concise yet thorough, clearly outlining a systematic approach. Limitations are clearly identified throughout the methods sections. However:

- The authors do not justify why they chose to puncture behind the cornea (102) and why 4% formalin fixation for 24 h (103) was chosen. Some reasoning around this may alleviate concerns of over-fixation, for example if this protocol has been published elsewhere or has been previously tested on other tissue.

The results section is detailed and well written, with frequent reference to figures which were overall informative. Testing for hypothyroidism was a convincing way of assessing whether the aardvark retinae were aberrant. Checking heterochromatin distribution to confirm photoreceptor identities was a nice touch, and the related discussion of semi-inverted heterochromatin contextualised these results. Overall, the discussion also communicated well, and the included subsections lend themselves well to supporting the results. The figures are mostly good, but some changes are warranted:

- Figure 2: Add proper scale bars to images. These would be more informative than estimating from graph paper in Figure 2C.

- Figure 5 A, C, H, J: Would improve legibility to include the target protein in the confocal images (i.e., “DAPI” and “rho4D2”), as done for ‘NeuN’ and ‘Gln synthase’ in Figure 4. At least in the first images where used. Then, representing stains in the same colour consistently will help legibility (e.g. instead varying between blue, grayscale, and red for DAPI).

- Figure 5B: the figure legend states ‘Clearly, the vast majority of ONL somata belong to rods.’ Authors should demonstrate the association between rhodopsin and DAPI by showing a panel with the ONL zoomed in. Moreover, pointing out where DAPI was not associated with rhodopsin would further emphasise this.

- Figure 6D: To better appreciate the localisation of ribbons and lack of co-localisation with S-opsin, an example image at higher magnification (or digital zoom) of the OPL would be beneficial.

- Including the some of the images used to determine photoreceptor densities (rods and cones) from different retinal regions would be instructive, for example by stitching together flat-mount sample fields (if contiguous), demonstrating periphery vs. central retina. Otherwise, a topographic schematic could summarise the same information.

Other minor adjustments:

- 246: There is no curvature estimation in Figure 2A. Perhaps intention was to demonstrate that the cornea had collapsed? If so, reference for Figure 2A should appear after ‘[…] because the cornea had collapsed’, not after ‘[…] its original curvature was estimated’.

- 294: Figure 2A, D are more informative than Figure 1 here.

- 297: The optic disk appears to be in a more nasal position on the temporal-nasal axis, as opposed to central.

- 309: Add ‘(CC)’ after ‘choriocapillaris’, as CC is used as an abbreviation later in the text.

- 331, 439, 522: The occasional deference to the discussion section is appropriate, although the consistent use of ‘(see Discussion)’ is preferred over statements like ‘This is addressed in the Discussion’.

- 391: Rh1 is implied, but text only states ‘rod opsin’. Given existence of Rh2 (although not in mammals), it would reduce ambiguity to specify that the rhodopsin is Rh1.

- 519: The shape of the ribbon is not easily discernible from Figure 6E2. A higher zoom would help support the assertion.

- 532: For more precise language, change ‘open’ to ‘unknown’ in ‘[…] it remains open whether PNA […]’.

- 596-598: The sentence about the repeated claim of aardvarks lacking cones can be excluded, as the point is made well-enough in the introduction.

- 621-623: The reference to a YouTube video is not needed, as Figure 1 is convincing.

- 640: The idea of a mixed tapetum type is rather quickly dismissed and would benefit from justifying why the authors think a mix is unlikely. A brief comment would suffice.

- 776: the authors should cite the specific works instead of ‘references therein’.

- 773-775: Would be easier to parse as ‘Aardvarks may use different sensitivities of pure L and S cones vs. dual pigment cones for colour vision’.

Overall, a very nicely presented assessment of the aardvark eye structure and photoreceptor identities. Looking forward to seeing the final publication, as well as the follow-up on inner-retina cell types.

6. PLOS authors have the option to publish the peer review history of their article (what does this mean? ). If published, this will include your full peer review and any attached files.

**Do you want your identity to be public for this peer review?** For information about this choice, including consent withdrawal, please see our Privacy Policy .

Reviewer #1: **Yes: ** Takeshi Yoshimatsu

Reviewer #2: **Yes: ** Simen Bruøygard

---

## [Author Response · Author response to Decision Letter 0]

29 Jan 2025

Our responses to the comments and suggestions of the editor and reviewers are detailed in the submitted Word file "Response to Reviewers".

---

## [Decision Letter · Decision Letter 1]

13 Feb 2025

Eye features and retinal photoreceptors of the nocturnal aardvark (Orycteropus afer, Tubulidentata)

PONE-D-24-50728R1

Dear Dr. Peichl,

We’re pleased to inform you that your manuscript has been judged scientifically suitable for publication and will be formally accepted for publication once it meets all outstanding technical requirements.

Kind regards,

Gerrit Hilgen

Academic Editor

PLOS ONE

Additional Editor Comments (optional):

Reviewers' comments:

Reviewer's Responses to Questions

**Comments to the Author**

1. If the authors have adequately addressed your comments raised in a previous round of review and you feel that this manuscript is now acceptable for publication, you may indicate that here to bypass the “Comments to the Author” section, enter your conflict of interest statement in the “Confidential to Editor” section, and submit your "Accept" recommendation.

Reviewer #1: All comments have been addressed

Reviewer #2: All comments have been addressed

2. Is the manuscript technically sound, and do the data support the conclusions?

Reviewer #1: Yes

Reviewer #2: Yes

3. Has the statistical analysis been performed appropriately and rigorously? 

Reviewer #1: N/A

Reviewer #2: Yes

4. Have the authors made all data underlying the findings in their manuscript fully available?

Reviewer #1: Yes

Reviewer #2: Yes

5. Is the manuscript presented in an intelligible fashion and written in standard English?

Reviewer #1: Yes

Reviewer #2: Yes

6. Review Comments to the Author

Reviewer #1: The authors address my concerns adequately. The manuscript provides a rare and valuable description of the eyes of the nocturnal aardvark.

Reviewer #2: We are pleased with the submitted revisions and think all suggestions have been either addressed in the edit or appropriately justified in the author's reply.

7. PLOS authors have the option to publish the peer review history of their article (what does this mean? ). If published, this will include your full peer review and any attached files.

**Do you want your identity to be public for this peer review?** For information about this choice, including consent withdrawal, please see our Privacy Policy .

Reviewer #1: **Yes: ** Takeshi Yoshimatsu

Reviewer #2: **Yes: ** Simen Bruoygard

---

## [Editor Report · Acceptance letter]

PONE-D-24-50728R1

PLOS ONE

Dear Dr. Peichl,

I'm pleased to inform you that your manuscript has been deemed suitable for publication in PLOS ONE. Congratulations! Your manuscript is now being handed over to our production team.

Kind regards,

on behalf of

Dr. Gerrit Hilgen

Academic Editor

PLOS ONE